# 5D operando tomographic diffraction imaging of a catalyst bed

A. Vamvakeros[1,2,3,4], S.D.M. Jacques[3], M. Di Michiel[4], D. Matras[2,5], V. Middelkoop[6], I.Z. Ismagilov[7], E.V. Matus[7], V.V. Kuznetsov[7], J. Drnec[4], P. Senecal[1,2] & A.M. Beale [1,2,3]

We report the results from the first 5D tomographic diffraction imaging experiment of a complex Ni–Pd/CeO$_2$–ZrO$_2$/Al$_2$O$_3$ catalyst used for methane reforming. This five-dimensional (three spatial, one scattering and one dimension to denote time/imposed state) approach enabled us to track the chemical evolution of many particles across the catalyst bed and relate these changes to the gas environment that the particles experience. Rietveld analysis of some $2 \times 10^6$ diffraction patterns allowed us to extract heterogeneities in the catalyst from the Å to the nm and to the μm scale (3D maps corresponding to unit cell lattice parameters, crystallite sizes and phase distribution maps respectively) under different chemical environments. We are able to capture the evolution of the Ni-containing species and gain a more complete insight into the multiple roles of the CeO$_2$-ZrO$_2$ promoters and the reasons behind the partial deactivation of the catalyst during partial oxidation of methane.

[1] Department of Chemistry, University College London, 20 Gordon Street, London WC1H 0AJ, UK. [2] Research Complex at Harwell, Rutherford Appleton Laboratory, Harwell Science and Innovation Campus, Harwell, Didcot OX11 0FA, UK. [3] Finden Limited, Merchant House, 5 East St. Helens Street, Abingdon OX14 5EG, UK. [4] ESRF, 71 Avenue des Martyrs, 38000 Grenoble, France. [5] School of Materials, University of Manchester, Manchester M13 9PL, UK. [6] Flemish Institute for Technological Research, VITO NV, Boeretang 200, 2400 Mol, Belgium. [7] Boreskov Institute of Catalysis SB RAS, Pr. Akademika Lavrentieva 5, Novosibirsk, Russian Federation 630090. Correspondence and requests for materials should be addressed to A.V. (email: antony@finden.co.uk) or to S.D.M.J. (email: simon@finden.co.uk) or to A.M.B. (email: andrew.beale@ucl.ac.uk)

Heterogeneous functional materials and devices, like catalytic solids, batteries and fuel cells tend to possess complex structures where the 3D spatial distribution of the various components is rarely uniform[1–3]. Such materials are known to change with time under operating conditions, and in order to gain an insight into the structure–function relationships, it is highly desirable to study them in situ with spatially resolved techniques[4–7]. Non-destructive X-ray spectroscopic/scattering techniques are typically employed to study such materials, but it is the brilliant X-rays generated at synchrotrons coupled with state-of-the-art detectors and tomographic data collections that now allow the acquisition of spatially resolved signals from within the interiors of intact objects under operating conditions[8–11]. Operando chemical imaging in 5D by synchrotron X-ray Diffraction-Computed Tomography (XRD-CT) could emerge as a game-changing technique for the non-destructive investigation of functional materials in space and time under real process conditions[12–14]. This chemical tomographic technique, along with the complementary pair distribution function computed tomography technique, has been employed in several cases to study catalysts and more recently batteries, under in situ/operando conditions[15–24]. Such multi-scale chemical imaging tools hold the potential to revolutionise our understanding of the relationships between structure and functionality in complex, real world, catalytic materials (ie, in particular to better differentiate between reactive and spectator species); information that can help us unravel the mechanisms underpinning catalytic reactions and catalyst deactivation and rationally design improved materials.

The catalytic partial oxidation of methane (POX) is considered a very promising alternative to the highly energy-demanding steam reforming of methane (highly endothermic reaction) to produce synthesis gas (CO and $H_2$) at gas-to-liquids (GTL) industrial plants (Fischer–Tropsch synthesis)[25–27]. The POX reaction is only mildly exothermic and it leads to a $H_2/CO$ molar ratio of 2, which is suitable for methanol and Fischer–Tropsch synthesis[28]. Given the promise of the POX reaction to decrease significantly the energy requirements of GTL plants, it is highly desirable to understand the spatio-temporal physico-chemical changes taking place in the real working catalyst. Such information is crucial in order to rationally design improved catalysts as the chemical (ie, gas composition) and temperature gradients in a reactor (both axially and radially) can have a direct impact on the chemical state of the catalyst[29–34].

Ni/$Al_2O_3$-based catalysts have been the most widely studied POX catalysts mainly due to Ni being a cheap material compared to the noble metals (eg, Pd, Pt, Rh and Ru). However, Ni/$Al_2O_3$-based catalysts are usually prone to deactivation with time under POX reaction conditions for a variety of reasons; the most often reported in literature being carbon deposition on active Ni sites (metallic Ni being the active catalyst component), sintering of Ni particles and solid-state reactions involving Ni (eg, the formation of $NiAl_2O_4$ in Ni/$Al_2O_3$ catalysts)[35,36]. The $CeO_2$–$ZrO_2$ support promoters can offer improved oxygen storage capacity and redox properties, enhanced metal-support interaction (higher dispersion and stability of the Ni species) and increased catalytic performance at lower temperatures, while small amounts of noble metal (Pd, Pt, Rh and Ru) can enhance the reducibility of the Ni species through a hydrogen spillover mechanism[37–41]. In this work, we employed the XRD-CT technique to investigate the behaviour of a complex 10 wt.% Ni–0.2 wt.% Pd/10 wt.% $CeO_2$–$ZrO_2$/$Al_2O_3$ catalyst under different operating conditions.

Until now, the XRD-CT temporal resolution has been considered its main drawback along with problems associated with large datasets and high volume processing. However, in this work, we implemented a new data collection strategy, the concept of which we introduced previously, in principle applicable to all pencil beam tomographic techniques, where both tomographic axes (ie, translation and rotation) are allowed to move simultaneously[42]. This new data collection strategy, coupled with the brilliant X-rays produced at the ESRF and the state-of-the-art Pilatus2M CdTe area detector allowed us to collect an XRD-CT dataset in <2 min (ie, 117 s); the data collection rate is at least one order of magnitude faster than that previously reported (see also Supplementary notes 1 and 2). Also, we were able to meet the data handling challenge by analysing millions of diffraction patterns.

## Results

**Multi-length scale physico-chemical imaging.** The in situ experiments were performed at beamlines ID31 and ID15A of the ESRF using the same catalyst (Supplementary Figures 1–3). The first experiment was a five-dimensional (5D) tomographic diffraction imaging experiment. Explicitly, here we mean three spatial dimensions, one diffraction dimension ($q$-space), and one dimension covering imposed chemical environment. Actually, one could consider this experiment covering more than five dimensions (ie, dependencies of multiple parameters such as phase distribution, crystallite size, and so on, over temperature and time too). The behaviour of a Ni–Pd/$CeO_2$–$ZrO_2$/$Al_2O_3$ catalyst was investigated during reduction and re-oxidation (redox experiment). In total, four 3D-XRD-CT datasets (each consisting of 30 XRD-CT datasets) were collected at the following operating conditions: ambient (1), 800 °C under He flow (2), 800 °C under 20% $H_2$/He flow (3) and 800 °C under 20% $O_2$/He flow (4). We present the results from the full Rietveld analysis of these datasets; the phase identification and Rietveld approach taken are described in the methods and Supplementary methods sections. The results represent the analysis of $\sim 2 \times 10^6$ diffraction patterns, which is an order magnitude larger compared to that previously reported[22,24,43–45].

The fresh catalyst is shown to contain crystalline NiO, PdO, $CeO_2$, $ZrO_2$ and $Al_2O_3$ (Supplementary Figure 4). Figure 1 presents the results from the XRD-CT dataset collected at the middle of the catalyst bed, the top row showing phase distribution maps of all the crystalline phases present in the catalyst particles. Each phase distribution map corresponds to the (normalised) values of the scale factors for each phase. The main catalyst support material, $Al_2O_3$, is seen to be homogeneously distributed over the catalyst particles and clearly defines the border and shape of each particle in the bed.

NiO is seen to be present in all catalyst particles as expected, due to the high Ni loading (10 wt.%), but its spatial distribution is not the same in all particles. We note, in the traditional 2D XRD-CT scan, it is not known whether the bottom, middle or top of each catalyst particle is probed, but as can be seen in Fig. 2, this is resolved in the 3D-XRD-CT; specifically, this figure shows the volume rendering of the scale factors of all phases. We see that NiO is found in higher concentration close to the surface of the catalyst particles. Similarly, PdO is also seen to be mainly present at/near the surface of the catalyst particles (this is clearly visible in Figs. 1 and 2), which is a direct outcome of the preparation method (impregnation) and the low Pd loading (0.2 wt.%). However, it can also be seen that there are regions where there is high concentration of PdO (hotspots in Fig. 1) indicating this phase is not well-dispersed over catalyst particles (see also Supplementary Figures 6–9).

The $CeO_2$ and $ZrO_2$ phases are most interesting. As is clear in Fig. 1, the $ZrO_2$ map shows an egg-shell distribution (only present at/close to the surface of the catalyst particles). The same phenomenon is observed in the $ZrO_2$ phase distribution volume shown in Fig. 2. Although one might expect the same distribution

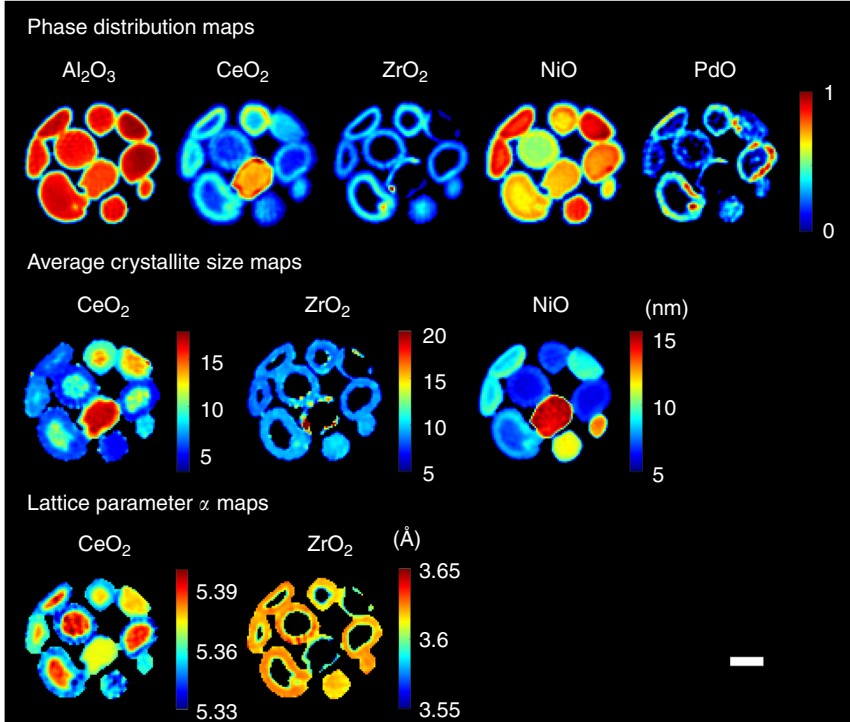

**Fig. 1** Maps derived from the Rietveld analysis of the XRD-CT data collected at the middle of the fresh catalyst bed. Top: phase distribution maps of $Al_2O_3$, $CeO_2$, $ZrO_2$, NiO and PdO corresponding to the normalised scale factors of these phases. Middle: average crystallite size maps of $CeO_2$, $ZrO_2$ and NiO (colorbar axes in nm). Bottom: Maps corresponding to the lattice parameter $a$ of $CeO_2$ and $ZrO_2$ unit cells (colorbar axes in Å). Scale bar corresponds to 0.5 mm

for the $CeO_2$ phase (due to co-impregnation method used for the preparation of the $CeO_2$–$ZrO_2$/$Al_2O_3$ support), that is not the case. There is a strong signal generated by the $CeO_2$ phase in the regions where $ZrO_2$ is also present, but it does not diminish closer to the centre of the catalyst particles (as the $ZrO_2$ signal does). As shown in Fig. 2, the observations regarding the $CeO_2$/$ZrO_2$ distributions are in full agreement with the rest of the XRD-CT datasets. The maps presented in the middle row of Fig. 1 correspond to values of the average crystallite size of the $CeO_2$, $ZrO_2$ and NiO phases, while the volume rendering of these data are presented in Fig. 2. It should be noted that the axial and radial gradients of these values (ie, spatial variations of the crystallite size values of each phase) provide the important physico-chemical information.

Characteristic examples are the crystallite size maps and volumes of the $CeO_2$ and $ZrO_2$ phases. The range of the $ZrO_2$ crystallite size is generally seen to be very narrow (ca. 8–12 nm), but in the regions where there are hotspots (areas of high concentration) of $ZrO_2$ (top row of Fig. 1), the average crystallite size increases significantly (middle row of Fig. 1). The same conclusion is reached by comparing the $ZrO_2$ volumes shown in Figs. 2 and 3, as the regions of high $ZrO_2$ concentration correspond to higher crystallite sizes. At first glance, one might think that the case of the $CeO_2$ phase is similar (ie, due to the co-impregnation method). For example, the catalyst particle rich in $CeO_2$ shown in Fig. 1 (approximately in the middle of the $CeO_2$ phase distribution map), also corresponds to highly crystalline $CeO_2$ (ie, highly crystalline compared to other regions of the sample). However, the range of $CeO_2$ crystallite sizes is large in the other catalyst particles. In fact, the values of the $CeO_2$ crystallite sizes are shown to follow an egg-yolk distribution. It can be seen in Fig. 1, that in the core of most particles, where

there is no $ZrO_2$, the $CeO_2$ crystallite size varies between 8 and 12 nm, while at/close to the catalyst surface, where there is $ZrO_2$ present, the $CeO_2$ crystallite size is less than half that value (ie, between 4 and 6 nm). This phenomenon is more apparent at the $CeO_2$ crystallite size volumes presented in Fig. 2, where it can be readily seen that the $CeO_2$ crystallite size follows an egg-yolk distribution in all catalyst particles.

The NiO crystallite size distribution shows a similar behaviour to that observed for the $ZrO_2$ phase. Explicitly, the NiO concentration is relatively higher close to the surface of the catalyst particles. In these regions, the average NiO crystallite size is also typically higher compared to the inner core of the catalyst particles (Fig. 1). However, it should be noted that there is a peculiar case too. The catalyst particle in the middle of the catalyst bed in Fig. 1 shows the highest values for NiO crystallite size, but the NiO concentration is not higher compared to the other catalyst particles. It seems that the chemistry of this specific particle may differ from the rest as there is also high concentration of $CeO_2$ (high with respect to the other catalyst particles). This may imply that in this particle there is a different chemical interaction between the various components (NiO, $CeO_2$ and $Al_2O_3$). It should be noted though, that no other crystalline phases were identified to be present in this particle.

The maps shown in the bottom row of Fig. 2 correspond to the refined values of lattice parameter $a$ of the $CeO_2$ and $ZrO_2$ unit cells. There are two distinct ranges regarding the values of the $ZrO_2$ lattice parameter $a$. More specifically, as shown in Figs. 1 and 3, in the regions of the sample where there are high concentrations of $ZrO_2$ (hotspots in Fig. 1 and regions of high intensity in Fig. 2), the values of the lattice parameter $a$ are low (<3.6 Å). In all other regions, where $CeO_2$ is present, the $ZrO_2$

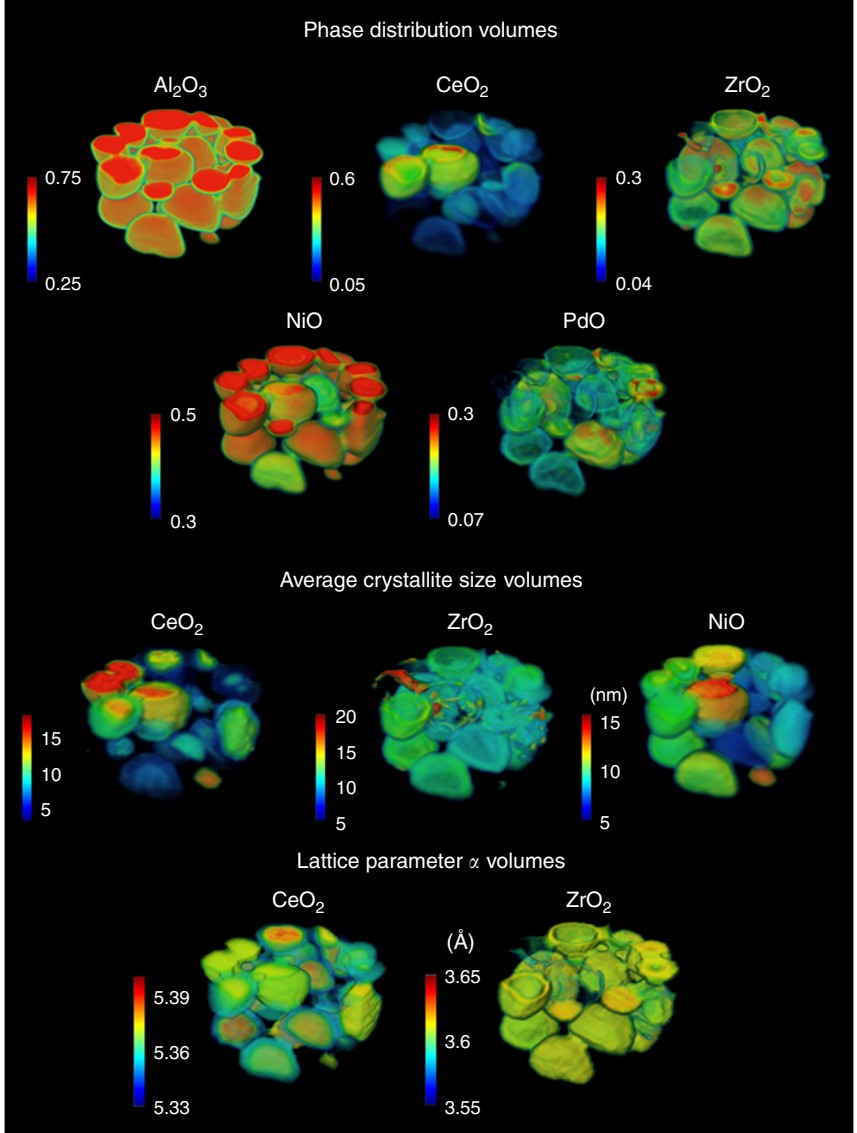

**Fig. 2** 3D maps obtained from the Rietveld analysis of the 3D-XRD-CT data collected at room temperature. Top: volume rendering of the normalised scale factors (phase distribution volumes). The values in the colorbar axes have been chosen to achieve the best possible contrast. Middle: volume rendering of the average crystallite size maps of $CeO_2$, $ZrO_2$ and NiO (colorbar axes in nm). Bottom: volume rendering of the lattice parameter $a$ of $CeO_2$ and $ZrO_2$ unit cells (colorbar axes in Å)

lattice parameter $a$ corresponds to higher values (>3.62 Å). This implies that there are actually two different $ZrO_2$ phases present in the sample: (1) a high-purity $ZrO_2$ present in areas of high concentration of $ZrO_2$ and (2) a Zr-rich $Ce_xZr_yO_2$ phase (Ce incorporation in the $ZrO_2$ unit cell leads to larger unit cell)[46–49]. Interestingly though, two distinct $CeO_2$ crystalline species seem to be present in the catalyst particles too. As shown in Figs. 1 and 3, the $CeO_2$ lattice parameter $a$ near the centre of the catalyst particles is notably higher (5.38–5.40 Å) compared to closer to the surface of the catalyst (5.34–5.36 Å), where $ZrO_2$ is present too. This implies that there is incorporation of a $Zr^{4+}$ substituent onto a $Ce^{4+}$ site in the $CeO_2$ unit cell in the regions where both $CeO_2$ and $ZrO_2$ are present leading to Ce-rich $Ce_xZr_yO_2$ phase (smaller unit cell compared to pure $CeO_2$)[37–40]. Summarising the $CeO_2$–$ZrO_2$ results derived from the Rietveld analysis of the 3D-XRD-CT data of the fresh catalyst, it can be concluded that there are four distinct crystalline $CeO_2$–$ZrO_2$ species present in the catalyst (see also Supplementary Figure 10):

1. Small crystallites of a Ce-rich $Ce_xZr_yO_2$ ($x \gg y$) phase near the surface of the catalyst particles;
2. Larger crystallites of a higher purity $CeO_2$ phase closer to the core of the catalyst particles;
3. Small crystallites of a Zr-rich $Ce_xZr_yO_2$ ($x \ll y$) phase near the catalyst surface;
4. Larger crystallites of a higher purity $ZrO_2$ phase where there is high concentration of $ZrO_2$ (near the surface of the catalyst particles—hotspots of this material).

**Catalyst activation and re-oxidation.** After the 3D-XRD-CT measurement was performed at ambient conditions, the temperature of the system was increased to 800 °C (temperature ramp rate of 20 °C per min) under the flow of He (volumetric flow rate of 100 ml min$^{-1}$). The catalyst remained at 800 °C under He flow (volumetric flow rate of 100 ml min$^{-1}$) for 1 h while collecting a 3D-XRD-CT scan. The gas mixture was then switched to a 20%

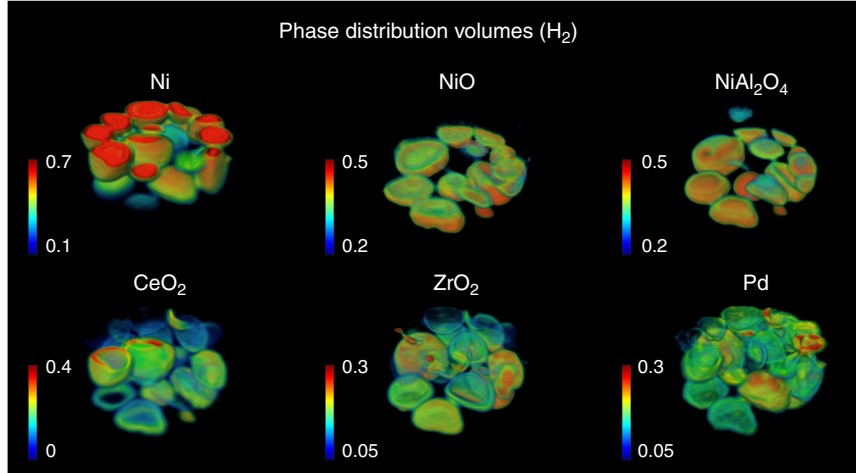

**Fig. 3** Catalyst activation process. Volume rendering of the normalised scale factors (phase distribution volumes) obtained from the Rietveld analysis of the 3D-XRD-CT data collected at 800 °C under 20% H$_2$/He flow. The values in the colorbar axes have been chosen to achieve the best possible contrast

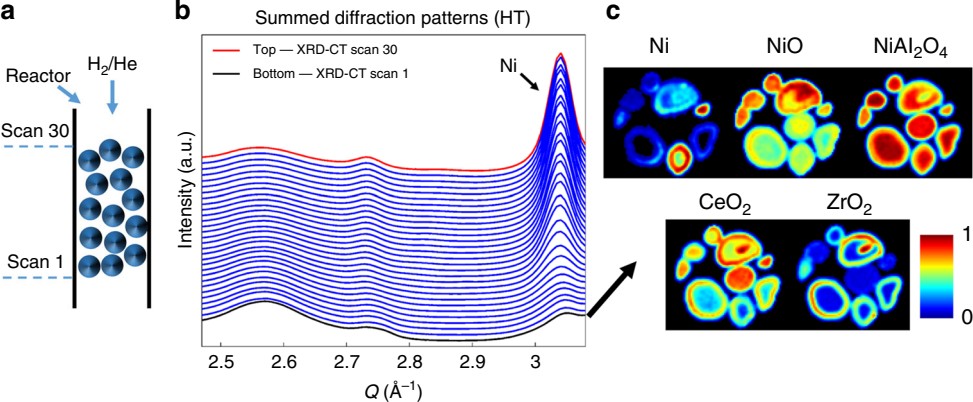

**Fig. 4** Reduction and activation of the catalyst bed. **a** Schematic representation illustrating the initial and final probing position during the 3D-XRD-CT at 800 °C under 20% H$_2$/He flow. The gases were introduced into the reactor from the top as indicated by the blue arrows. **b** The summed diffraction patterns of the 3D-XRD-CT data (30 XRD-CT scans). The black arrow indicates the highest intensity Ni peak. **c** Phase distribution maps of Ni, NiO, NiAl$_2$O$_4$, CeO$_2$ and ZrO$_2$ corresponding to the normalised scale factors of these phases as derived from the Rietveld analysis of XRD-CT scan 1 as indicated by the black arrow

H$_2$/He (total flow rate of 100 ml min$^{-1}$). Another 3D-XRD-CT measurement was then performed under these reducing conditions in an attempt to capture the state of the activated catalyst. Finally, the gas mixture was switched to 20% O$_2$/He (total flow rate of 100 ml min$^{-1}$) and a last 3D-XRD-CT scan was performed (catalyst re-oxidation experiment).

The main difference in the state of the catalyst at 800 °C under He flow compared to the room temperature 3D-XRD-CT data is related to the Ni-containing phases. Specifically, we observe the presence of the undesired NiAl$_2$O$_4$ phase (Supplementary Figures 11 and 12). It has been previously reported in literature that crystalline NiAl$_2$O$_4$ can be observed in Ni/Al$_2$O$_3$ catalysts calcined at temperatures above 600 °C[50]. It is perhaps not surprising then that the NiAl$_2$O$_4$ phase is seen to be present in the catalyst at 800 °C[51]. This indicates that a reducing chemical environment (eg, H$_2$) is essential in order to avoid the NiO to NiAl$_2$O$_4$ transition and reduce the NiO to the desired, active metallic Ni phase, as the formation and growth of the NiAl$_2$O$_4$ phase is a temperature-driven phenomenon taking place even under inert chemical environment (He flow). However, the phase

distribution maps of NiO, NiAl$_2$O$_4$ and Ce$_x$Zr$_y$O$_2$ revealed that NiO is mainly present near the surface of the catalyst particles (where the most of the Ce$_x$Zr$_y$O$_2$ phases are present) while NiAl$_2$O$_4$ is closer to their core. This provides direct evidence that one of the roles of the Ce$_x$Zr$_y$O$_2$ phases is to stabilise the NiO phase and suppress the formation of the undesired NiAl$_2$O$_4$ phase (Supplementary Figure 12).

The results from the Rietveld analysis of the 3D-XRD-CT data collected during reduction (catalyst activation) are presented in Fig. 3. In contrast to the results obtained under He flow, there are significant differences between the 30 XRD-CT 'slices' consisting this 3D-XRD-CT dataset. This is clearly shown at the right side of Fig. 4 where the summed diffraction patterns from all 30 XRD-CT datasets collected during the 3D-XRD-CT measurement under reducing conditions are presented. The diffraction peak that appears at ca. $Q = 3.04$ Å$^{-1}$ corresponds to the metallic Ni phase (reflection (111)). The crystalline metallic Ni phase is seen to form quickly and is present even in the first XRD-CT data collected under H$_2$ flow (bottom of the sample volume probed— XRD-CT scan 1). However, the continuous growth of the Ni

diffraction signal and decrease of the NiO and NiAl$_2$O$_4$ peaks (ca. $Q = 3$ Å$^{-1}$ and $Q = 2.55$–$2.65$ Å$^{-1}$, respectively) imply that NiO and NiAl$_2$O$_4$ were still present during the acquisition of this 3D-XRD-CT measurement (ie, in several XRD-CT scans). In order to decouple these phenomena, we performed another diffraction experiment where we show that the NiO/NiAl$_2$O$_4$/Ni concentration gradients shown in Fig. 4 are a purely temporal phenomenon (Supplementary Figures 13–15 and accompanying text). The retardation in the full reduction of the Ni–O and Ni–Al–O

species could be due to the formation/presence of water produced from the formation of metallic Ni closer to the reactor inlet[19,23].

The phase distribution volumes shown in Fig. 3 clearly demonstrate this Ni concentration gradient along the length of the catalyst bed. Similarly, the decrease of the concentration of the NiO and NiAl$_2$O$_4$ phases along the catalyst bed can also be readily observed. A closer inspection of the results also reveals that the NiAl$_2$O$_4$ phase, which is known to be more difficult to reduce compared to the NiO phase, remains present at positions of the catalyst bed where the NiO diffraction signal has already diminished[25,35]. Of course, as also shown in the 30 XRD-CT-summed diffraction patterns presented in Fig. 4, only metallic Ni is observed near the top of the catalyst bed.

The Ce$_x$Zr$_y$O$_2$ mixed oxides seem to be the predominant phases compared to the respective higher purity CeO$_2$ and ZrO$_2$ phases. This is implied from the CeO$_2$ phase distribution volume shown in Fig. 3 where CeO$_2$ is seen to be mainly present near the surface of the catalyst particles, similar to the ZrO$_2$ (ie, Zr-rich Ce$_x$Zr$_y$O$_2$) phase distribution volume (egg-shell distribution). It should be emphasised that the most important result obtained from this 3D-XRD-CT dataset is related to the correlation between the Ni and the Ce$_x$Zr$_y$O$_2$ phase distribution maps. Specifically, as shown on the right side of Fig. 4, the metallic Ni phase is first formed in the regions (initially following only an egg-shell distribution before being present everywhere in all catalyst particles) where the Ce$_x$Zr$_y$O$_2$ phases are present. This result provides direct evidence that the most important role of the Ce$_x$Zr$_y$O$_2$ promoters is to enhance the reducibility of the NiO and the formation of the active catalyst component, the metallic Ni phase.

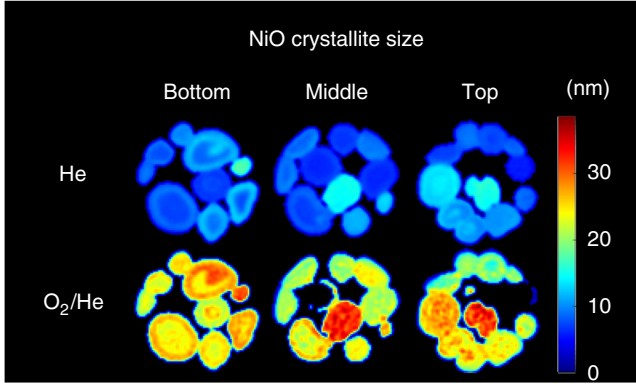

**Fig. 5** NiO crystallite size evolution. Maps of crystallite sizes of NiO (units in nm) as obtained from the Rietveld analysis of the XRD-CT data collected at the bottom, middle and top of the sample (XRD-CT scans 1, 15 and 30, respectively). Top row: catalyst under He flow, bottom row: catalyst under 20% O$_2$/He flow. Scale bar corresponds to 1 mm

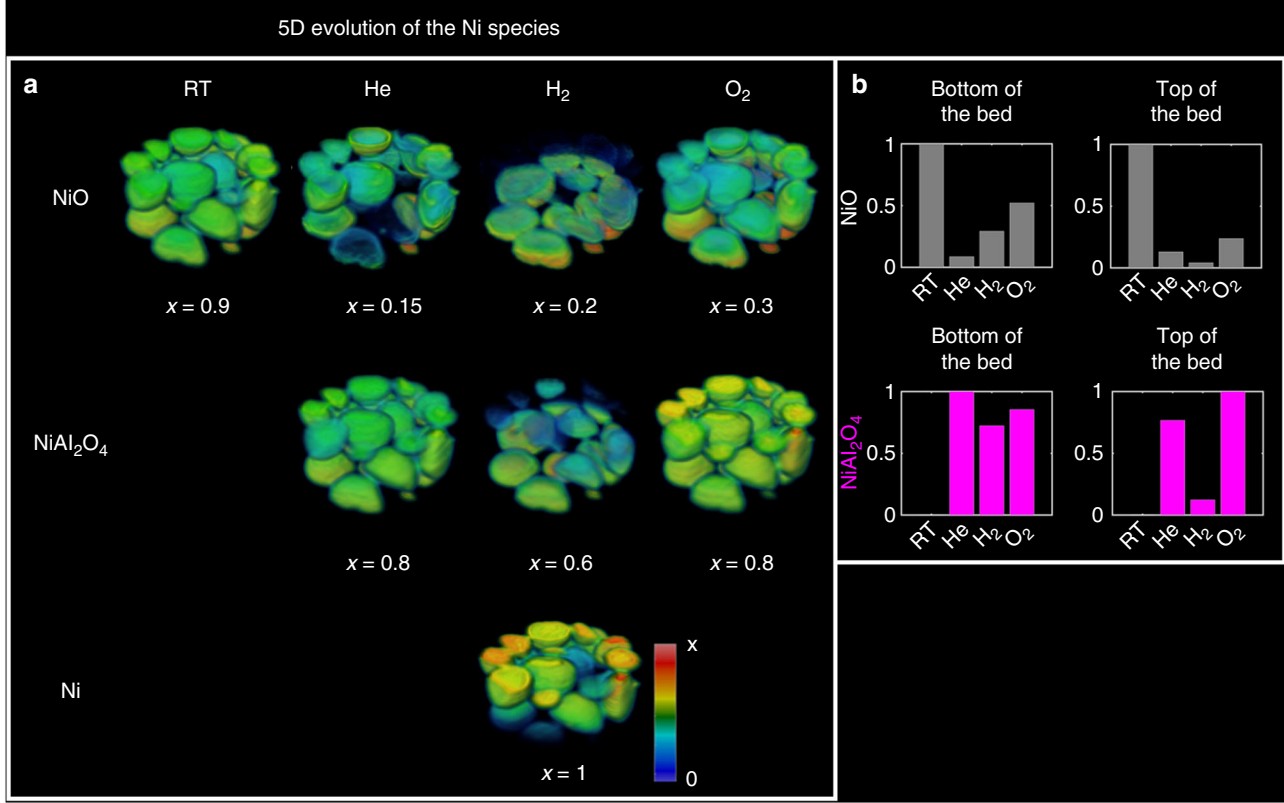

**Fig. 6** 5D chemical evolution of the Ni-containing species in the catalyst bed. **a** Phase distribution volumes of NiO, NiAl$_2$O$_4$ and Ni as obtained from the Rietveld analysis of the 3D-XRD-CT data collected at the four different operating conditions. The values in the colorbar axes have been chosen to achieve the best possible contrast. **b** Solid-state evolution of the NiO and NiAl$_2$O$_4$ phases at the bottom and top of the catalyst bed during the redox experiment (ie, XRD-CT scan 1 and 30). The results presented in this figure correspond to the Rietveld analysis of ca. $1.2 \times 10^6$ diffraction patterns

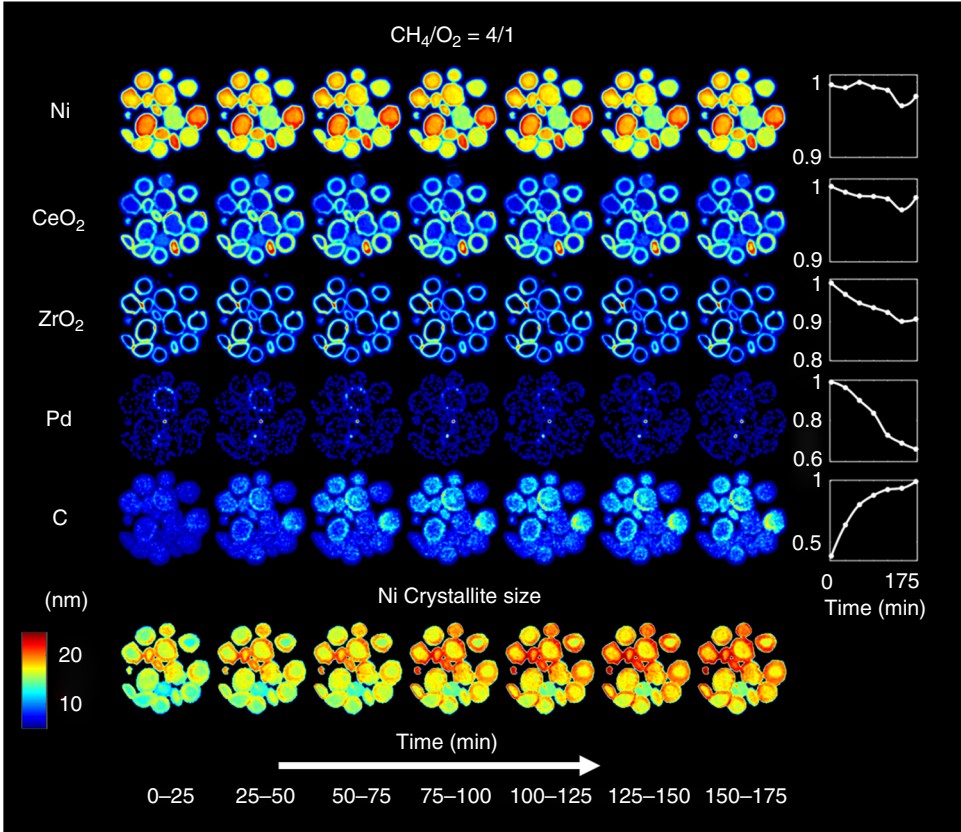

**Fig. 7** Chemical evolution of the catalyst during partial oxidation of methane. Phase distribution maps of Ni, CeO$_2$, ZrO$_2$, Pd and C (graphite) corresponding to the normalised scale factors of these phases derived from the Rietveld analysis of the XRD-CT data collected during the POX experiment. Bottom: maps of crystallite sizes of Ni under POX reaction conditions (colour bar axes in nm). Scale bar corresponds to 1 mm

Finally, it should be mentioned that metallic Pd is observed at the 3D-XRD-CT data collected under reducing conditions. The phase distribution volume of Pd shown in Fig. 3 is very well correlated with the PdO distribution presented previously in Fig. 2. It is also important to note that the metallic Pd peak (reflection (111)) is observed at larger $Q$ values than expected (ca. $Q = 2.85\,\text{Å}^{-1}$) corresponding to a smaller unit cell than a pure Pd one (Supplementary Figure 17). This result implies that Ni is incorporated in the Pd unit cell (shift of diffraction peak to larger $Q$ values) and that Pd exists as a Ni$_x$Pd$_y$ phase (alloy) rather than a high-purity Pd phase; the formation of Ni$_x$Pd$_y$ alloys has been previously reported in other catalytic systems[52–55].

The last 3D-XRD-CT dataset was collected under the flow of 20% O$_2$/He. The re-oxidation process of metallic Ni to NiO is really fast as the metallic Ni peak is absent in all 30 XRD-CT datasets and only the NiO and NiAl$_2$O$_4$ phases can be observed (Supplementary Figure 16). The most important results from this re-oxidation step are presented in Fig. 5 where the maps corresponding to the crystallite size of NiO, as obtained from the Rietveld analysis of the XRD-CT data collected at the bottom, middle and top of the sample under He and under 20% O$_2$/He flow, are shown. It can be readily seen that the re-oxidation step at 800 °C led to a substantial increase in the crystallite size of the NiO phase, implying that sintering of the NiO species took place. In many regions of the sample, the NiO crystallite size more than doubled from ca. 5–15 nm to ca. 20–35 nm. The increase of the NiO crystallite size was not prevented even in the regions where the various CeO$_2$/ZrO$_2$/Ce$_x$Zr$_y$O$_2$ promoters are present indicating that careful consideration must be given to the catalyst pre-treatment before switching to real reaction conditions.

The evolution of the Ni-containing crystalline phases during this 5D tomographic diffraction imaging redox experiment is summarised in Fig. 6 where the phase distribution volumes of NiO, NiAl$_2$O$_4$ and Ni under the various operating conditions are displayed (panel a). As discussed previously, NiO is the only observed crystalline phase in the fresh catalyst, while NiAl$_2$O$_4$ is the predominant Ni-containing phase in the 3D-XRD-CT dataset collected at 800 °C under He flow. Under these conditions, the NiO follows mainly an egg-shell distribution, similar to the Ce$_x$Zr$_y$O$_2$ phase (where $y > x$), while the NiAl$_2$O$_4$ phase is mainly present closer to the core of the catalyst particles. This information provides direct evidence that the Ce$_x$Zr$_y$O$_2$ suppresses/delays the formation of the undesired NiAl$_2$O$_4$ phase.

As discussed previously, under reducing conditions, a Ni chemical gradient is observed along the length of the bed, which is a time effect (ie, delay in the full reduction of NiO and NiAl$_2$O$_4$). This is also illustrated in panel b where it is shown that first the NiO is reduced and then the NiAl$_2$O$_4$ phase (probing from bottom to top of the bed). A similar time effect is shown during the re-oxidation experiment as the top of the bed (last scan) contains less NiO compared to the bottom of the bed (first scan) indicating that the NiO to NiAl$_2$O$_4$ transition can also be captured.

**Partial oxidation of methane**. The last experiment was an operando XRD-CT experiment of the same catalyst during the partial oxidation of methane. The same protocol was used for the pre-treatment of the catalyst as in the redox experiment. Specifically, the temperature of the system was increased under the flow of He up to 800 °C (temperature ramp rate of 20 °C per min)

and then the catalyst was activated under the flow of 20% $H_2$/He (total flow rate of 100 ml min$^{-1}$) to form the metallic Ni phase. The catalyst bed was then exposed to a POX reaction mixture (ie, 90 ml min$^{-1}$ of pure He, 12 ml min$^{-1}$ of pure $CH_4$ and 3 ml min$^{-1}$ of pure $O_2$—$CH_4$/$O_2$ molar ratio of 4:1), which was kept constant for the duration of the experiment and seven XRD-CT scans were collected in total.

In Fig. 7, the phase distribution maps of certain crystalline phases of interest, as obtained from the Rietveld analysis of the XRD-CT data collected at 800 °C under POX reaction conditions, are presented. These phase distribution maps correspond to the values of the scale factors for each phase normalised with respect to the maximum values. NiO and $NiAl_2O_4$ remain only in traces for the duration of the experiment, while $Al_2O_3$ is not changing; for these reasons, these maps are not shown here and emphasis is given on the active catalyst components. More importantly, the crystalline Ni phase is seen to be very stable for the duration of the POX experiment (ca. 3 h). We also note there is no evidence for formation of crystalline Ni carbide phases (eg, $Ni_3C$ or NiC)[56].

The typical reaction mixture used to test the performance of POX catalysts is a gas mixture where the $CH_4$/$O_2$ molar ratio is 2:1 (stoichiometric ratio for the POX reaction)[57–59]. Here, a higher $CH_4$/$O_2$ molar ratio of 4:1 was used in an attempt to force the catalyst to deactivate faster and capture the corresponding solid-state changes. As it is shown in Fig. 7, the Ni remains in a metallic form for the duration of the experiment (ca. 3 h), which is an interesting result especially when one considers the methane-rich operating conditions. It should also be stated that no other crystalline Ni-containing phases were observed in the XRD-CT data beyond the trace amounts of NiO and $NiAl_2O_4$ mentioned previously. However, as is shown in the bottom part of Fig. 7, sintering of metallic Ni is seen to take place as a function of time during the POX experiment.

Figure 7 also tells us that the diffraction signal from both $Ce_xZr_yO_2$ phases is seen to decrease with time under reaction conditions although one may argue that the decrease is not very significant (ca. 10 % decrease after 3 h). The Pd phase (more precisely, the $Ni_xPd_y$ alloy as discussed previously) is also seen to decrease as a function of time under the POX reaction conditions. It should be noted though that no other crystalline Pd-containing species were observed in the XRD-CT data collected during the POX experiment. The mass spectrometry data acquired during the POX experiment are presented in Supplementary Figure 18, serve to prove that the catalyst was captured in its active state during the POX reaction. No apparent deactivation of the catalyst was observed during the experiment, but as it is shown in Fig. 7, upon switching to the POX reaction mixture a new phase formed, which was identified as graphite (Supplementary Figure 17)[55]. From the phase distribution maps presented in Fig. 7, it can be clearly seen that the Ni and graphite maps are directly correlated. Specifically, the results suggest that the coke deposition initially takes place at the regions rich in Ni, which is the active catalyst component for the POX reaction. This graphite phase is seen to continuously grow during the POX reaction experiment. It is also implied that particles that are rich in $Ce_xZr_{1-x}O_2$ are less prone to graphite formation; in these particles, the $Ce_xZr_{1-x}O_2$ forms a protective layer near the surface of the catalyst particles. As mentioned previously, the POX reaction experiment lasted for ~3 h and it can be seen that the diffraction signal from this phase doubled during this time period (Fig. 7). It is reasonable to expect that the growth of graphite at the catalyst particles will probably influence the long-term performance of the catalyst (ie, by blocking the active Ni/Pd sites). This suggestion seems also to be consistent with laboratory catalytic activity measurements we have performed following the same catalyst pre-treatment and activation protocol (Supplementary Figures 19 and 20).

Specifically, the catalyst showed high performance when it was exposed to POX reaction conditions (the $CH_4$/$O_2$ molar ratio of 2:1), leading to 84% $CH_4$ conversion and almost 100% $H_2$ yield. However, exposing the catalyst to the $CH_4$/$O_2$ molar ratio equal to 4:1 for 3 h, partially deactivated the catalyst as it did not regain its initial performance when the reaction mixture was switched back to 2:1. As shown by the in situ XRD-CT data, this is highly likely to be due to the graphite formation (see also Supplementary Figures 21–26).

## Discussion

It is well known that time-resolved in situ or operando X-ray spectroscopic or scattering studies provide information on evolving structure activity relationships in functional materials with researchers in the field of heterogeneous catalytic materials shown historically to be keen to exploit their potential[60]. In more recent times, traditional single-point measurements have been superseded (in terms of the information they provide) by X-ray chemical imaging studies and these have been shown to yield far greater insight into the structure of single particles or else a handful of particles in a cross-section[1,2,5,11,17,20,21]. Despite these important advantages, there will always be doubts as to whether a single particle/group of particles are truly representative of their ilk in terms of composition and/or behaviour (ie, in a reactor environment). The biggest challenge in this regard is to be able to follow changes in the catalyst particle structure as a function of their location in a reactor. It is well known that catalyst particles experience variation in reactant/product composition both radially and longitudinally, thereby rendering attempts to correlate structure–function relationships in a reactor difficult and in some cases impossible. The 3D chemical imaging results presented herein obtained in real time and under operando conditions therefore represent significant step in attempting to extract more meaningful correlations between structure and function in catalytic and other functional materials.

Specific to this study, we have shown how the Rietveld analysis of the reconstructed XRD-CT data allowed us to differentiate between four different $Ce_xZr_yO_2$ crystalline species present in the sample. This information could only be obtained by collecting the 3D-XRD-CT dataset and performing the Rietveld analysis. It should be emphasised that the analysis of the reconstructed data ($>2 \times 10^6$ diffraction patterns) although very challenging, is now possible on reasonable timescales so as to obtain unprecedented physico-chemical information regarding the state of the catalyst under different operating conditions. We were able to create, for the first time, 3D maps corresponding to phase distributions, crystallite sizes and unit cell lattice parameters, demonstrating the multi-length scale information that can be obtained from performing such an experiment and the corresponding data analysis. As such, we were able to effectively track the evolution of the crystalline Ni-containing species under different chemical environments (He, $H_2$/He, $O_2$/He, $CH_4$/$O_2$). NiO was the only crystalline Ni-containing phase present in the fresh catalyst and it was shown that a reducing environment is necessary in order to avoid the formation of the undesired $NiAl_2O_4$ phase at high temperatures. It was shown that the role of the $Ce_xZr_yO_2$ promoters is to suppress this conversion and enhance the reduction of the Ni oxides to the catalytically active metallic Ni. It should be emphasised that this chemical information could have not been obtained with any conventional technique but only from the spatially resolved diffraction signals present in an XRD-CT dataset.

The reduction process was seen to be significantly slower compared to the re-oxidation. Specifically, significant Ni/NiO/

$NiAl_2O_4$ concentration gradients were observed along the catalyst bed during reduction but not during re-oxidation. This phenomenon was attributed to the water formation during the reduction of the $NiO/NiAl_2O_4$ species to metallic Ni. It was also shown that the re-oxidation of the catalyst at 800 °C under 20% $O_2$/He flow can have a strong impact on the NiO crystallite size. Metallic Ni was seen to be the main crystalline Ni-containing phase of the catalyst under POX reaction conditions, but sintering of the Ni crystallites took place as a function of time under reaction conditions. More importantly though, crystalline graphite was seen to form and continuously grow under POX reaction conditions which, as the laboratory catalytic activity data collected under the same conditions indicate, caused the partial deactivation of the catalyst.

We have shown that it is possible to track the chemical evolution of many particles across a catalyst bed and relate these changes and variations to the non-uniform gas composition the catalyst particles experience. The insights and understanding gained from this in situ study can be used to rationally design improved methane-reforming catalysts. With the advancements in synchrotron brightness, detector performance, sample environment (new reactor cells) and data analysis (Rietveld analysis of XRD-CT data), multi-dimensional chemical imaging techniques are bound to become increasingly easier to perform and one can readily foresee that they will replace conventional in situ XRD and X-ray imaging as the preferred method for characterising functional materials and devices (eg, catalytic reactors, batteries and fuel cells).

## Methods

**Catalyst preparation**. The 10 wt.% Ni–0.2 wt.% Pd/10 wt.% $CeO_2$–$ZrO_2$/$Al_2O_3$ catalyst was prepared by sequential impregnation method. The $CeO_2$–$ZrO_2$/$Al_2O_3$ support was prepared by the co-impregnation method[61,62]. The microspherical (γ + δ)-$Al_2O_3$ with granules size of ~500 µm was used. Approximately 500 µm of (γ + δ)-$Al_2O_3$ was impregnated by aqueous solution of salts (cerium nitrate $Ce(NO)_3$ · $6H_2O$ and oxychloride of zirconium ZrOCl · $8H_2O$) at the required ratio. The $CeO_2$–$ZrO_2$/$Al_2O_3$ was dried at 120 °C for 6 h and calcined under air at 850 °C for 6 h with a heating rate of 2 °C per min. The 10 wt.% $CeO_2$–$ZrO_2$/$Al_2O_3$ support was impregnated by aqueous solution of nickel nitrate salt $Ni(NO_3)_2$ · $6H_2O$ of the appropriate concentration. Then, Ni/$CeO_2$–$ZrO_2$/$Al_2O_3$ was dried at 120 °C for 6 h and calcined in air at 500 °C for 4 h with a heating rate of 2 °C per min. The Ni/$CeO_2$–$ZrO_2$/$Al_2O_3$ was impregnated by aqueous solution of palladium nitrate Pd $(NO_3)_2$ salt of the appropriate concentration. The catalyst were then dried at 120 °C for 6 h and calcined in air at 500 °C for 4 h with a heating rate of 2 °C per min. The catalyst samples used in this study were kindly provided by the Boreskov Institute of Catalysis (BIC).

**Reactor cells**. The three catalytic reactors investigated in this study consisted of 10 wt.%Ni–0.2 wt. Pd/10 wt.% $CeO_2$–$ZrO_2$/$Al_2O_3$ catalysts (ie, quartz capillary fixed bed reactors supported by glass wool); the catalyst loading was 35 mg for the operando experiment (4 mm outer diameter quartz capillary for the operando experiment, 2 mm for the 5D tomographic diffraction imaging experiment). In each case, the reactor was mounted into a gas delivery stub, itself mounted to a standard goniometer (to enable alignment). The goniometer was fixed to a rotation stage set upon a translation stage to facilitate the movements required for the CT measurement. At ID31, heating was achieved by virtue of two Cyberstar hot air blowers heating each side of the catalytic reactor while at ID15A a furnace designed for CT experiments was used. Temperature calibration was performed before all experiments using a thermocouple by measuring the temperature at the catalyst bed. During the operando XRD-CT measurements, the outflow gasses were monitored by mass spectrometry using an Ecosys portable mass spectrometer. The mass spec line was inserted inside the capillary from the top. The mass spec capillary is dragging with a constant flow rate of 20 ml min$^{-1}$.

**5D tomographic diffraction imaging measurements at ID31, ESRF**. XRD-CT measurements were made at beamline station ID31 of the ESRF using a 70 keV monochromatic X-ray beam focused to have a spot size of 25 × 25 µm. 2D powder diffraction patterns were collected also using the state-of-the-art Pilatus3 X CdTe 2 M hybrid photon counting area detector. The total acquisition time per point was 15 ms (exposure time of 11 ms and readout time of 4 ms). Four 3D-XRD-CT scans of the Ni–Pd/$CeO_2$–$ZrO_2$/$Al_2O_3$ catalyst were acquired at different operating conditions: (1) at room temperature, (2) at 800 °C under He flow, (3) at 800 °C

under 20% $H_2$/He flow (reduction step) and (4) at 800 °C under 20% $O_2$/He flow (re-oxidation step). Each 3D-XRD-CT scan composed of 30 XRD-CT scans, each one collected at a different vertical position (ie, 25 µm step size along the catalyst bed). Each XRD-CT scan lasted ~2 min. The tomographic measurements were made with 100 translation steps (translation step size of 25 µm) covering 0–180° angular range, in steps of 2.5° (ie, 72 line scans). The detector calibration was performed using a $CeO_2$ NIST standard. Every 2D diffraction image was converted to a 1D powder diffraction pattern after applying an appropriate filter (ie, 1% trimmed mean filter) to remove outliers using in-house developed MATLAB scripts[63]. The final XRD-CT images (ie, reconstructed data volume) were reconstructed using the filtered back projection algorithm.

**In situ XRD mapping and XRD-CT measurements at ID15A, ESRF**. XRD mapping and XRD-CT measurements were made at beamline station ID15A of the ESRF using a 91 keV monochromatic X-ray beam focused to have a spot size of 40 × 20 µm (horizontal × vertical). 2D powder diffraction patterns were also collected also using the state-of-the-art Pilatus3 X CdTe 2 M hybrid photon counting area detector. The total acquisition time per point was 10 ms. Initially, an XRD-CT scan was performed at the middle of the catalyst bed at ambient conditions. The XRD-CT scan was made with 151 translation steps (translation step size of 40 µm) covering 0–180° angular range, in steps of 1.44° (ie, 126 line scans). Calibration and processing (5% trimmed mean filter) was carried as described in the previous section (5D imaging measurements at ID31, ESRF) with X-ray detection again using the Pilatus3 X detector. The temperature of the system was increased under the flow of pure He (30 ml min$^{-1}$) up to 800 °C with a ramp rate of 20 °C per min. An XRD-CT scan was collected and then the inlet gas mixture was switched to 20% $H_2$/He (50 ml min$^{-1}$). Eight successive XRD maps were collected covering the whole bed (151 translation steps and 12 z positions, each 0.5 mm apart). Each XRD map lasted ~5 min. Finally, a 3D-XRD-CT scan was collected after the XRD maps. The 3D-XRD-CT scan composed of 10 XRD-CT scans, each one collected at a different vertical position (ie, 80 µm step size along the catalyst bed).

**Operando XRD-CT measurements at ID31, ESRF**. XRD-CT measurements were made at beamline station ID31 of the ESRF using a 70 keV monochromatic X-ray beam focused to have a spot size of 20 × 20 µm. Here, the total acquisition time per point was 20 ms. Tomographic measurements were made with 225 translation steps (translation step size of 20 µm) covering 0–180° angular range, in steps of 1.125° (ie, 160 line scans). Calibration and processing was carried as described in the section above (5D imaging measurements at ID31, ESRF) with X-ray detection again using the Pilatus3 X detector. The temperature of the reactor was then increased to 800 °C with a ramp rate of 20 °C per min under the flow of He (ie, 30 ml min$^{-1}$). The inlet gas was then switched to a 20% $H_2$/He gas mixture (total flow rate of 100 ml min$^{-1}$). Finally, the catalyst bed was exposed to a POX reaction mixture (ie, 90 ml min$^{-1}$ of He, 12 ml min$^{-1}$ of $CH_4$ and 3 ml min$^{-1}$ of $O_2$ having a $CH_4$/$O_2$ molar ratio of 4:1), which was kept constant for the duration of the experiment.

**Rietveld analysis of the XRD-CT data**. Quantitative Rietveld refinement was performed using the reconstructed diffraction patterns using the TOPAS software, on a voxel by voxel basis[64]. The results from the refinements were imported into MALTAB in order to create the various figures presented in this work (eg, phase distribution maps based on the scale factors or weight percentages, lattice parameters, and so on). Unless stated otherwise, the Rietveld analysis of the XRD-CT data presented in this work was based on the intensity of the scale factors and should be treated as a semi-quantitative analysis. Rietveld analysis was performed using the summed diffraction pattern of each XRD-CT dataset prior to the Rietveld analysis of the XRD-CT data in order to have a good starting model before performing the batch Rietveld analysis.

## Data availability

Copies of raw radially integrated XRD-CT data can be found at http://tiny.cc/NCOMMS18-12413A. All data are available from the corresponding authors on reasonable request.

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

## Acknowledgements

The development of the catalyst used in this work was funded within the DEMCAMER project as part of the European Union Seventh Framework Programme (FP7/2007–2013) under grant agreement no. NMP3-LA-2011–262840. Note: 'The present publications reflect only the authors' views and the Union is not liable for any use that may be made of the information contained therein'. The authors would like to thank the ESRF for beamtime. Antonis Vamvakeros and Dorota Matras (respectfully in full and in part) are supported through funding received from the European Union Horizon 2020 research and innovation programme under grant agreement no. 679933 (MEMERE project). Andrew M. Beale acknowledges EPSRC (grant EP/K007467/1) for an Early Career Fellowship. The authors would like to thank Thomas Buslaps and Denis Duran for their help during the experimental setups at beamline ID15A of the ESRF and Gavin Vaughan for optimising the continuous rotation-translation XRD-CT data acquisition macro.

## Author contributions

The 5D redox and POX experiments were conceived by A.V., V.M., S.D.M.J. and A.M.B. M.D.M. and A.V. implemented the continuous rotation-translation data collection strategy, wrote the macros for the data acquisition and the scripts for the sinogram data pre-processing. A.V., P.S., D.M., S.D.M.J. and A.M.B. performed the 5D redox and POX experiments. M.D.M. and J.D. were responsible for ID15A and ID31 instrumentation and setup at the ESRF, respectively. The high-resolution XRD-CT experiments was conceived and performed by A.V. and M.D.M. A.V. and D.M. performed the micro-CT and 3D-XRD-CT comparison experiment. Data processing and analysis was performed by A.V. with assistance from M.D.M., S.D.M.J., D.M. and A.M.B. I.Z.I., E.V.M. and V.V.K. performed the catalyst synthesis/preparation and the laboratory catalytic activity experiments. V.M. was responsible for the Scanning Electron Microscopy (SEM)/Energy Dispersive X-ray Spectrometry (EDS) sample preparation and measurement. D.M. was responsible for the Thermogravimetric Analysis (TGA) sample preparation and measurement. A.V., S.D.M.J. and A.M.B. are responsible for writing the manuscript with feedback given by all contributors. A.M.B. directed the research.

## Additional information

**Competing interests:** The authors declare no competing interests.

