## [Peer Review File · Nature Communications]

Reviewer #1 (Remarks to the Author):

This paper reports operando imaging combining tomography and XRD and the authors investigated the Ni-Pd/CeO₂-ZrO₂/Al₂O₃ catalysts under various conditions. The 3D imaging combining X-ray analytical technique is now state-of-art but I cannot accept this paper by the following reasons. After revision by the authors, the manuscript would be suitable for Angew. Chem. or J. Catal.

- (1) The methodology of imaging technique is already reported although there is improvement of resolution in this report. The authors visualized the detail of each element but discussion from the imaging data can be predictable from previous studies.
- (2) The "5-dimensional" imaging is not clear, this is one of the keywords of this paper and the authors should explain it well at the beginning of the paper.
- (3) Scale bars and axes are necessary in all 2D maps and volume renderings (Figs. 1-7 and ESI). Are the particles visualized in the paper single particles or the secondary aggregation of single particles? The comparison with SEM or TEM images for the same observation field should be presented in the manuscript.
- (4) There are too many figures with similar information in the manuscript (the authors include 8 figures in the manuscript!). The time scales is missing in Fig. 8 (Fig. S6 too) and it is difficult to discuss the detail of reaction phenomena from the present figures.
- (5) It is well known that the activity of solid catalysts highly depends on preparation. The catalytic performace data of their catalyst under steady-state conditions should be added and compared to commercial or conventional catalysts for the reaction.
- (6) Compared to Ni, Pd can be well distributed on catalyst support. The distirbtion of PdO was better than that of NiO in Figure 1 but there were several aggregation parts on the catalyst. I cannot calculate the size of the PdO phases because of the lack of scale bar in the figure. Considering the quite low loading of Pd (0.2 wt%), it looks too large for practical catalysts. Related to Q5, it is necessary to evaluate the practical activity of the catalyst. How about STEM-EDS analysis of the catalyst, does it agree to the imaging results?
- (7) Carbon deposition is one of the most serious reasons of deactivation of supported Ni catalyst. C mapping in Fig. 8 (missing reaction time) shows serious C deposition proceeded on the catalyst. It is very important where carbon deposition for catalyst activation accelerated, the structural information related to the carbon deposition is quite helpful for catalyst design. The authors should be analyze it from the imaging data.
- (8) There were significant formation of NiAl₂O₄ in the samples. Along to the reaction profile in Fig.8, the ratio and location of the several Ni phases should be discussed in the viewpoint of structure-activity relationship.

Reviewer #2 (Remarks to the Author):

5D tomographic operando diffraction imaging of a catalyst bed

Vamvakeros et al.

This paper describes partial oxidation of methane (POX) to synthesis gas over multicomponent Ni-Pd/CeO₂-ZrO₂/Al₂O₃ catalysts. This is without doubt an important reaction that gains more importance as the fossil resources decrease and also methane in small quantities should be used. Furthermore, it plays an important role in the frame of solid oxide fuel cells.

On the other hand the paper deals in an impressive way with the rapid acquisition and deconvolution of complex 3D tomographic diffraction data acquired under various oxidizing, reducing and reaction conditions, in order to observe crystalline phase structure in the catalyst and relate this to the experimental conditions (mainly changes in gas environment). This is of high relevance.

The piece of information, which the manuscript describes, is therefore potentially excellent and to my opinion a powerful example of the rapidly developing potential of synchrotron diffraction tomography applied to catalysis. While the method itself is not exactly new, this manuscript is clearly an advance over the authors' previous publications in this field, in terms of data quality, structural observations and overall presentation. Aside from the many synchrotron measurement campaigns this must have involved, processing of such data is on its own a formidable task and the authors should be commended on this.

However, the study should be much better presented. While the structural observations and the core of the results seem reasonable, I believe the work has some inconsistencies and at points the experimental method is unclear. Furthermore, this work will surely stimulate discussion about the precise definition of an 'in situ' vs an 'operando' experiment (as the title claims). I therefore recommend major revisions for the manuscript to be refined before publication. Specific comments are listed below.

Specific points:

1) Title: Why is 5D used in the title and why is this not explained in the abstract. In principle you receive 3D-maps and some more information as function of time etc. Concerning “operando”, see further below.

2) Content: More emphasis should be given to the core result which is Figure 7 and Figure 8.

3) Literature: There are spatially resolved studies that report the structure of a catalyst during catalytic partial oxidation of methane that should be cited as well as modeling studies that simulate concentration and temperature gradients in reactors during this reaction in 3D.

4) Method: One of the central topics and stated advances of this work is streamlining the acquisition of XRD-tomography data, and reducing the scan times required (line 65-71). While there is little doubt the authors managed to achieve such impressive speeds, this is due to a rather unusual and intriguing data collection strategy cited in reference 36 - an unpublished work. I have strong reservations about referencing unpublished works, particularly when they are directly relevant to the experiments performed here. I would recommend all efforts be made to publish reference 36 before this manuscript is published or make all the information available in the manuscript.

5) Emphasis of synchrotron: The distinction between ‘synchrotron X-ray tomography’ and ‘X-ray tomography’ is perhaps not clear to the casual reader. The fact is that the work presented here is not possible outside the synchrotron - and this fact should be somewhat more apparent. It would be misleading for the reader to think that such data could be acquired in a lab source, although modern sources are very effective for absorption contrast even down to $\mu\text{m-nm}$ scale.

For curiosity, why was it necessary to perform measurements at two different beamlines ID31 and ID15A? It appears essentially the same beam parameters and detectors were used in both cases. What was the difference between the two beamlines regarding data acquisition and experimental methods offered?

6) Experimental: The description of the reaction cells used is rather lacking in detail, and in the current form is probably not sufficient for most readers to understand the complexity of performing these measurements (e.g. specific geometry, positioning of gas blowers, mass spec, free rotation requirements). It is reasonable to assume that the cells used are the same as described in previous works by the authors [Ref 15 - O'Brien et al, *Chemical Science* 3, 509 (2012); Ref 19 - Senecal et al, *ACS Catalysis* 7, 2284-2293 (2017)] - while a full description of the apparatus does not need to be repeated here, more appropriate citation in the experimental section would be appreciated. With this in mind, the publication of this manuscript is very likely to stimulate interest among the catalysis community to attempt similar experiments, likely with the same apparatus which is presumably

available at ID31/ID15A. Hence, please provide an objective assessment of the drawbacks of the reaction cell used.

Specifically:

(i) it is not a 'closed' system, the top of the reactor is open to the environment, and backflow of ambient gas into the capillary can only be countered by supplying rather high flow rates of gas (in this work 100 ml/min during POX experiments).

(ii) the space velocities of the gases inserted into the system are therefore necessarily rather high. It would be useful to state the GHSV applied during tomography experiments, and how this compares to the laboratory experiments.

7) Catalytic performance: The claim of an operando experiment suggested in the title rests entirely on the mass spectrometry data obtained, and specifically the observation of products (CO + H₂, byproducts CO₂, H₂O). This data is found only in the supporting information and is not completely convincing, for the following reasons:

(i) the t = 0 point is given as more or less the exact moment the gas conditions were changed from pre-reducing (20% H₂/He, 100 ml/min) to POX reaction conditions (30:4:1 He:CH₄:O₂, 105 ml/min).

(ii) The relative change in CO and H₂ mass traces before t = 0 (proof of product formation) is therefore difficult to see. The authors should clearly show the traces before POX conditions were introduced, so during the reduction step immediately prior.

(iii) Note further that the total flow rate of gas to the MS was not constant at this point (100 to 105 ml/min). The signal of ALL gases detected by the MS will therefore change.

(iv) the appearance of CO₂ in particular is rather apparent, but not totally convinced the POX reaction was occurring.

Linked to this previous point, the authors state in the supporting info (line 217-221) that all mass traces observed were stable during tomography studies, therefore catalyst deactivation was not expected. In Figure S13 I cannot see the trace of oxygen. I do not understand why CO₂ is zero and why the traces of water are not given. How does this reconcile with the gradual appearance of graphite, which is a likely product of combustion? Etc.

8) Deepening the point of operando:

Since the word 'operando' appears in the title and abstract, this must be carefully re-considered. By (generally accepted) definition, an in situ study involves spectroscopic/microscopic/physical probing under operating conditions, while an operando study requires catalytic activity to be measured simultaneously so that structure-activity relations can be derived. Apart from the catalytic data discussed previously I have the following reservations:

(i) operando studies need to be done on a closed system - the tomography capillary setup used here is not a closed system, but open at the top.

(ii) while there may be some evidence of product formation (see discussion on MS above), there is no attempt at quantification, while even qualitative discussion of the MS data is relegated to the supplementary information.

To justify the use of the term 'operando', the catalytic activity data presented should be robust - otherwise this is an 'in situ' study with 'realistic' reaction conditions.

9) Figures: While the many interesting figures are enjoyable and well presented, emphasis should be laid on Figures 7 and 8. One may put some of them into the ESI. Figure 8 should be improved. I assume moving from left to right is the same slice measured at different times. Therefore please introduce a time scale on the x axis for the top part of the figure (what is the temporal resolution for these scans). Furthermore, the small graphs on the right have no axis labels. Please include labels on all figure axes. In the discussion on Figure 8 (line 313-318), the authors state that the decreasing intensity of the CeZrO₂ phases (10% in 3 hours) is 'not very significant', although without time scales on the graphs it is hardly to judge. The Pd intensity seems to change rather dramatically, could the authors comment on this?

Some minor comments:

- line 15, '5D' should be defined as '5-dimensional' on first use, this is not a common acronym

- line 19-20, Angstrom and nm should have the word 'scale'

- line 33, "it is the brilliant X-rays generated at synchrotrons coupled with state of the art detectors and tomographic data collections that now allow acquisition of spatially-resolved signals under operating conditions". Not necessarily - it is very possible (99% of all such papers) to do spatially-resolved studies in 2D. Tomography is not required as this statement suggests (although 3D spatial resolution is always better than 2D). Please rewrite.

- line 43-44, "This is especially true for catalysts applied at the industrial scale where catalysts are needed to be produced on a scale where fine chemical control (and therefore sample homogeneity) is difficult to achieve". This sentence is difficult to read and the meaning is unclear.

- line 65-66, "...XRD-CT temporal resolution has been considered its main drawback." What about the huge volumes of data obtained? Data storage and particularly big data processing is becoming a more relevant issue in modern catalysis and may to be mentioned.

- line 76-78, the discussion of many dimensions (beyond 5D) is somewhat speculative and probably not necessary.

- Methods - Catalyst preparation, were the catalysts pressed and sieved after synthesis? More information is required - this is also relevant to calculate whether the GHSV of the POX reaction was as expected for standard operating conditions. Without catalyst bed volume, the reader cannot derive the GHSV.

- Methods - Reactor cells, line 412, where exactly was the temperature measured? Top of the bed, bottom of the bed? Is it right to assume the temperature was not continually measured during reaction? How can we then be sure that the stated temperatures were actually experienced by the catalyst during in situ / operando studies, since there is no feedback control mechanism to maintain a steady temperature (e.g. modulations due to exothermicity)?

- Methods, line 416, 'tomographic' is misspelled.

- line 437, "An XRD-CT was at the middle of the catalyst bed was collected at ambient conditions." Please rewrite.

- line 443, 'successive' is misspelled.

- line 443-444, "Eight successive XRD maps were collected covering the whole bed..." This implies that the authors actually scanned from top to bottom every single part of the catalyst bed. With a 0.5 mm vertical step size this is surely not the case? Rather selected slices must have been scanned?

- line 462, problem with reference formatting.

I recommend the manuscript to Nature Communications, if the authors can properly answer the comments and improve the manuscript as suggested above.

Reviewer #3 (Remarks to the Author):

Tracking the chemical evolution of catalysts under reaction conditions is important for better understanding of the structure-function relationships. In this manuscript, a novel 5D tomographic diffraction imaging technique was applied in practical chemical experiment (POM), which enabled the authors to extract heterogeneities in the catalyst under different chemical environments. The results are quite interesting. However, some key issues concerning this technique should be emphasized and additional data need to be provided. I would recommend that publication is withheld until following information is supplied.

1. It is generally accepted that the nanometer size of supported metal catalysts is closely related with its catalytic performance. As for XRD-CT technique, the limit of crystallite size resolution should be given in the manuscript, since the size of Pd/PdO is not showed (Figure 1).

2. The components of supported catalyst are complicated. The signals from different phases or metals may overlap with each other and affect the accuracy of data analysis. The spatial resolution of phase distribution maps should be emphasized.
3. The CeO₂-ZrO₂/Al₂O₃ support was prepared by co-impregnation method. It is reasonable to imagine that the Ce/Zr species would load on the surface of Al₂O₃ particle. However, obvious CeO₂ signal was found at the core of Al₂O₃ particle. The inherent reasons should be discussed (or only due to overlap of 3D signal?) .
4. In this manuscript, the chemical evolution of Ni species was detailed studied while that of Pd species, another active component, was seldom investigated. The authors should explain the role of Pd in the POM reactions.
5. The TG experiment should be conducted for the catalysts after POM reaction (3h) to further confirm the amount of coke.
6. As the author indicated, the 5D tomographic operando diffraction imaging showed high spatial and temporal resolution with good element sensitivity, a more detailed discussion on the coke deposition process (where and when it starts? On Ni or Pd particle or interface?) should be provided to strengthen the confidence on this novel technique.
7. The construction of structure-function relationship not only lies in recognition of the particle size or phase composition but also the electronic property of active species. The additional information from other characterization techniques such as XPS or XAFS except the 5D tomographic diffraction imaging were hoped to be given and the connection of these results can be discussed.
8. The figure caption should be more intelligible. Taking Figure 8 as an example, the meaning of different color and time schedule of different pictures should be provided.

The comments submitted by the reviewers have been addressed with the following changes where necessary:

Reviewer 1:

This paper reports operando imaging combining tomography and XRD and the authors investigated the Ni-Pd/CeO₂-ZrO₂/Al₂O₃ catalysts under various conditions. The 3D imaging combining X-ray analytical technique is now state-of-art but I cannot accept this paper by the following reasons. After revision by the authors, the manuscript would be suitable for Angew. Chem. or J. Catal.

(1) The methodology of imaging technique is already reported although there is improvement of resolution in this report. The authors visualized the detail of each element but discussion from the imaging data can be predictable from previous studies.

It is not clear to us what the reviewer means by saying “there is improvement of resolution in this report. The authors visualized the detail of each element but discussion from the imaging data can be predictable from previous studies”. We clearly state in the introduction part of the paper:

“Until now, the XRD-CT temporal resolution has been considered its main drawback. However, in this work, we implemented a new data collection strategy, in principle applicable to all “pencil” beam tomographic techniques, where both tomographic axes (i.e. translation and rotation) are allowed to move simultaneously. This new data collection strategy, coupled with the brilliant X-rays produced at the ESRF and the state-of-the-art Pilatus2M CdTe area detector allowed us to collect an XRD-CT dataset in less than 2 minutes (i.e. 117 s); the data collection rate is at least one order of magnitude faster than that previously reported.”

The reviewer has not clarified what they mean by resolution - clearly this is not true for spatial resolution. As reviewer 2 indicates, it is not trivial to simply improve time resolution and it cannot simply be obvious/predictable as to what the benefits of this would be. Obtaining operando data is key here and the observations in meaningful time-frames is the basis for performing such experiments. So to clarify to this reviewer, we report a major improvement in temporal resolution, not spatial, from our previous work. We have also provided a table in the ESI where we compare the data acquisition rates achieved in this work compared to other studies showing that we have made a major step towards real time 3D chemical imaging. We can readily foresee that the continuous rotation/translation data collection strategy presented in this work will become the standard strategy in the future not just for XRD-CT but for all “pencil beam” chemical tomographic techniques (e.g. XAFS/XRF/XRD/PDF/SAXS-CT).

We also cannot see how our previous work on chemical imaging and catalysis leads to this statement by the reviewer: “The authors visualized the detail of each element but discussion from the imaging data can be predictable from previous studies.” This is a new materials system that we have never investigated in the past and all the results are new. This 5D imaging experiment allowed us to get unprecedented physico-chemical information about this

complex materials system and track the evolving solid-state chemistry under different operating conditions.

(2) The "5-dimensional" imaging is not clear, this is one of the keywords of this paper and the authors should explain it well at the beginning of the paper.

We have now defined the 5D term in the abstract as requested by the reviewer. For the interest of clarity, we have added a new section in the ESI explaining multi-dimensional chemical imaging as we think this may prove very useful to the readers in general. We should emphasize though that the 5D tomographic diffraction imaging is clearly demonstrated in the supporting video entitled "5D_imaging.mp4". We understand that in many cases videos/animations added to the electronic versions of papers do not add much scientific value to the work presented and it seems like the reviewers did not look at this video (probably for that reason). From our personal experience it is easier to explain to non-experts/new users how these chemical CT techniques work using animations/videos rather than just static images. For this purpose, we put a lot of effort in creating this video, which is used to explain the 5D imaging.

(3) Scale bars and axes are necessary in all 2D maps and volume renderings (Figs. 1-7 and ESI). Are the particles visualized in the paper single particles or the secondary aggregation of single particles? The comparison with SEM or TEM images for the same observation field should be presented in the manuscript.

This is also spotted by the other reviewers and it is indeed very important, as such we have added scale bars to Figures 1, 6, 8, S12 and S14. We have also performed SEM and EDX analysis of both the fresh and spent POX catalyst. We have added this new section in the ESI. These results serve to support the results obtained from the XRD-CT experiments.

(4) There are too many figures with similar information in the manuscript (the authors include 8 figures in the manuscript!). The time scales is missing in Fig. 8 (Fig. S6 too) and it is difficult to discuss the detail of reaction phenomena from the present figures.

We disagree that there are too many Figures in the main paper - we believe that these are all necessary to understand the work presented in the paper. Nature Communications supports up to 10 figures. We spent a lot of time making sure we only included in the main paper only those figures that were necessary to provide a flowing and comprehensible story. By the nature of multidimensional data, covering spatial, chemical and temporal/environmental conditions, it is not always easy to present material that is readily accessible to non-expert readers. Critical information to the materials under study, such as phase distribution, crystallite size and lattice parameters should naturally be included in such a study. We have added a time scale to Figure 8 as requested by the reviewer.

(5) It is well known that the activity of solid catalysts highly depends on preparation. The catalytic performance data of their catalyst under steady-state conditions should be added and compared to commercial or conventional catalysts for the reaction.

To the best of our knowledge, the POX process has not been commercialised so we cannot see how the reviewer's comment is relevant. Furthermore, in this work we present catalytic activity data for the Ni-Pd/CeO₂-ZrO₂/Al₂O₃ catalyst which show that its performance is amongst the best according to the literature (we have also referenced in the paper all the

important review papers regarding POX catalysts). We would also like to comment here though that even if there was a commercial catalyst available, such a comparison would be beyond the scope of this work. The aim of this work was not to find the optimal conditions under which the catalyst will show the best result. Here, we investigated a complex multi-component catalytic system under real reaction conditions with 3D chemical imaging and were able to capture the solid-state chemical evolution of the various catalyst components. The catalytic performance data are provided in the ESI and as shown in review POX papers (mention which ones) already cited in this work, the performance of the catalyst is among the highest reported.

(6) Compared to Ni, Pd can be well distributed on catalyst support. The distribution of PdO was better than that of NiO in Figure 1 but there were several aggregation parts on the catalyst. I cannot calculate the size of the PdO phases because of the lack of scale bar in the figure. Considering the quite low loading of Pd (0.2 wt%), it looks too large for practical catalysts. Related to Q5, it is necessary to evaluate the practical activity of the catalyst. How about STEM-EDS analysis of the catalyst, does it agree to the imaging results?

Here, we disagree with the reviewer. The Pd is not well distributed (i.e. not homogeneously distributed as indicated by Figures 1 and 8 but also from Figure S8 in the ESI). As discussed in the paper, the weight loading of Ni is 10 wt. % while that of Pd is 0.2 wt. %. As expected, the NiO will be present in all regions in the catalyst particles. The PdO phase distribution maps show where the PdO is present, not how much or what is the crystallite size. The catalyst showed very good performance; we have provided catalytic activity data (Figure S20 in the ESI) showing that the performance of the catalyst is among the highest reported (POX review papers are referenced in the paper). The reality is that the preparation method will dictate the spatial distributions but despite a relatively simple procedure there will be several factors at play such as the acidity of the catalysts, the local pH during wetting and drying steps, the local concentration of components etc. It would indeed be an interesting study to look at how different preparation protocols affect the distribution of the components and try and relate these to performance. We had to limit our study and focussed alone upon the best performing catalyst. Finally, we want to emphasize that it is with this type of studies (i.e. by using chemical tomographic techniques), we can get an indication of how we can improve the catalyst design.

(7) Carbon deposition is one of the most serious reasons of deactivation of supported Ni catalyst. C mapping in Fig. 8 (missing reaction time) shows serious C deposition proceeded on the catalyst. It is very important where carbon deposition for catalyst activation accelerated, the structural information related to the carbon deposition is quite helpful for catalyst design. The authors should be analyze it from the imaging data.

We completely agree with the reviewer's comment and indeed we added more text regarding that in the main text of the paper. In short, we observe the Ni and graphite phase distribution maps are directly related. We performed TGA measurement of the spent catalyst which also confirm the presence of a single type of coke present in the catalyst (Figure S25). This is, though, a first experiment (it is a communication paper after all) and we report the first ever XRD-CT data of a working POX catalyst. We hope that we will have the chance in future beamtime experiments to investigate the catalyst under POX conditions focusing only on the coke formation. However, in this work we also showed, for the first time, what is the state of

the activated catalyst, after re-oxidation, the solid-state evolution of the crystalline Ni-containing phases (i.e. NiAl₂O₄/NiO/Ni) during redox and the multiple roles of the CeO₂-ZrO₂ promoters. These are all new findings and we expect that the catalysis community will appreciate them.

(8) There were significant formation of NiAl₂O₄ in the samples. Along to the reaction profile in Fig.8, the ratio and location of the several Ni phases should be discussed in the viewpoint of structure-activity relationship.

We clearly mention in the main paper: “In Figure 8, the phase distribution maps of all crystalline phases, as obtained from the Rietveld analysis of the XRD-CT data collected at 800 °C under POX reaction conditions, are presented. These phase distribution maps correspond to the values of the scale factors for each phase normalised with respect to the maximum values. This is an important detail as it might be implied from Figure 8 that the NiO and NiAl₂O₄ phases are present in large amounts under POX reaction conditions but this is not the case here (i.e. these phases remain only as traces under reaction conditions).” The NiAl₂O₄ and NiO remain only as traces for the duration of the POX experiment. As such and in order to avoid confusion we have removed them from Figure 8 and also changed the corresponding text in the main paper.

Reviewer 2:

This paper describes partial oxidation of methane (POX) to synthesis gas over multicomponent Ni-Pd/CeO₂-ZrO₂/Al₂O₃ catalysts. This is without doubt an important reaction that gains more importance as the fossil resources decrease and also methane in small quantities should be used. Furthermore, it plays an important role in the frame of solid oxide fuel cells.

On the other hand the paper deals in an impressive way with the rapid acquisition and deconvolution of complex 3D tomographic diffraction data acquired under various oxidizing, reducing and reaction conditions, in order to observe crystalline phase structure in the catalyst and relate this to the experimental conditions (mainly changes in gas environment). This is of high relevance.

The piece of information, which the manuscript describes, is therefore potentially excellent and to my opinion a powerful example of the rapidly developing potential of synchrotron diffraction tomography applied to catalysis. While the method itself is not exactly new, this manuscript is clearly an advance over the authors' previous publications in this field, in terms of data quality, structural observations and overall presentation. Aside from the many synchrotron measurement campaigns this must have involved, processing of such data is on its own a formidable task and the authors should be commended on this.

However, the study should be much better presented. While the structural observations and the core of the results seem reasonable, I believe the work has some inconsistencies and at points the experimental method is unclear. Furthermore, this work will surely stimulate discussion about the precise definition of an ‘in situ’ vs an ‘operando’ experiment (as the title claims). I therefore recommend major revisions for the manuscript to be refined before publication. Specific comments are listed below.

We thank the reviewer for his/her kind comments regarding our work and also for acknowledging the environmental and economic importance of the catalytic system under study. We also believe the high relevance of this study and that we demonstrate how synchrotron XRD-CT can be an invaluable tool to characterise not only working heterogeneous catalysts but functional materials in general. We are also glad that the reviewer is able to appreciate the amount of work put into analysing the very large amount of diffraction data collected during the XRD-CT experiments presented in this work.

Specific points:

1) Title: Why is 5D used in the title and why is this not explained in the abstract. In principle you receive 3D-maps and some more information as function of time etc. Concerning “operando”, see further below.

We agree with the reviewer and we added this in the abstract. It is also clearly stated later in the main paper: ‘The first experiment was a five-dimensional (5D) tomographic diffraction imaging experiment. Explicitly here we mean three spatial dimensions, one diffraction dimension (q-space), and one dimension covering imposed chemical environment. Actually, one could consider this experiment covering more than five dimensions (i.e. dependencies of multiple parameters such as phase distribution, crystallite size etc over temperature and time too).’

For the interest of clarity, we have taken the initiative and added a new section in the ESI explaining multi-dimensional chemical imaging as we think this may prove very useful to readers in general.

2) Content: More emphasis should be given to the core result which is Figure 7 and Figure 8.

We understand that these are the two most impressive figures presented in the paper but we believe the other figures contain very significant physico-chemical information about the system under study too. Figure 7 is in fact a summary of everything we have been describing in the main text up to that point as it also contains data presented in Figures 2, 4 and 5. We added more text explaining Figure 8 in the main text of the paper. In short, we observe the Ni and graphite phase distribution maps are directly related. We performed TGA measurement of the spent catalyst which also confirm the presence of a single type of coke present in the catalyst (Figure S25). We hope that in the future we will have the opportunity to perform an in situ 3D-XRD-CT experiment focusing only on the role of the different catalyst component during POX and gain an even better understanding of the structure-function relationships.

3) Literature: There are spatially resolved studies that report the structure of a catalyst during catalytic partial oxidation of methane that should be cited as well as modeling studies that simulate concentration and temperature gradients in reactors during this reaction in 3D.

We thank the reviewer for noticing this. To the best of our knowledge there are no such studies for Ni/Al₂O₃-based catalytic systems. We have added though the following relevant papers and very nice studies regarding noble metal catalysts for POX and a recent review paper:

Horn R, Williams KA, Degenstein NJ, Schmidt LD. Syngas by catalytic partial oxidation of methane on rhodium: Mechanistic conclusions from spatially resolved measurements and numerical simulations. *Journal of Catalysis* **242**, 92-102 (2006).

Kimmerle B, *et al.* Visualizing a Catalyst at Work during the Ignition of the Catalytic Partial Oxidation of Methane. *The Journal of Physical Chemistry C* **113**, 3037-3040 (2009).

Grunwaldt J-D, *et al.* Catalysts at work: From integral to spatially resolved X-ray absorption spectroscopy. *Catalysis Today* **145**, 267-278 (2009).

Korup O, *et al.* Catalytic partial oxidation of methane on platinum investigated by spatial reactor profiles, spatially resolved spectroscopy, and microkinetic modeling. *Journal of Catalysis* **297**, 1-16 (2013).

Hettel M, Diehm C, Torkashvand B, Deutschmann O. Critical evaluation of in situ probe techniques for catalytic honeycomb monoliths. *Catalysis Today* **216**, 2-10 (2013).

Morgan K, *et al.* Evolution and Enabling Capabilities of Spatially Resolved Techniques for the Characterization of Heterogeneously Catalyzed Reactions. *ACS Catalysis* **6**, 1356-1381 (2016).

4) Method: One of the central topics and stated advances of this work is streamlining the acquisition of XRD-tomography data, and reducing the scan times required (line 65-71). While there is little doubt the authors managed to achieve such impressive speeds, this is due to a rather unusual and intriguing data collection strategy cited in reference 36 - an unpublished work. I have strong reservations about referencing unpublished works, particularly when they are directly relevant to the experiments performed here. I would recommend all efforts be made to publish reference 36 before this manuscript is published or make all the information available in the manuscript.

We understand the reviewer's concern and we have now written in the main paper: "However, in this work, we implemented a new data collection strategy, the concept of which we introduced previously, in principle applicable to all "pencil" beam tomographic techniques, where both tomographic axes (i.e. translation and rotation) are allowed to move simultaneously; the technical details will be described in detail elsewhere." We cite also the relevant paper:

Vamvakeros A, *et al.* Interlaced X-ray diffraction computed tomography. *Journal of Applied Crystallography* **49**, 485-496 (2016).

In that work, we introduced the concept of this data collection strategy but we first implemented in this work here. The interested readers can refer to our previous work. The manuscript under preparation will be purely technical as it is designed to be a technical guide to such data collection strategies and as such it is beyond the scope of the current work to discuss these technicalities. We should mention though that the method is currently available (macros) for all users at ID15A and in the future to other ESRF beamlines.

Also, we took the initiative and added a short literature review in the ESI ("Fast XRD-CT and comparison with literature") comparing the acquisition times achieved in this work with previous work. This is done to directly show the major improvements we have made and how this is expected to become the new "standard" data collection strategy in chemical CT techniques.

5) Emphasis of synchrotron: The distinction between 'synchrotron X-ray tomography' and 'X-ray tomography' is perhaps not clear to the casual reader. The fact is that the work presented here is not possible outside the synchrotron - and this fact should be somewhat more apparent. It would be misleading for the reader to think that such data could be acquired

in a lab source, although modern sources are very effective for absorption contrast even down to $\mu\text{m-nm}$ scale.

For curiosity, why was it necessary to perform measurements at two different beamlines ID31 and ID15A? It appears essentially the same beam parameters and detectors were used in both cases. What was the difference between the two beamlines regarding data acquisition and experimental methods offered?

We have added the word **synchrotron** in the first paragraph but it clearly implied in two paragraphs in the introduction that this is synchrotron and not lab XRD-CT (this issue has not been raised by any other reviewer). In the first paragraph of the paper we clearly state:

“Non-destructive X-ray spectroscopic/scattering techniques are typically employed to study such materials but it is the brilliant X-rays generated at synchrotrons coupled with state-of-the-art detectors and tomographic data collections that now allow the acquisition of spatially-resolved signals under operating conditions. Operando chemical imaging in 5D by **synchrotron** XRD-CT could emerge as a game-changing technique for the non-destructive investigation of functional materials in space and time under real process conditions.”

In the last paragraph before the Results section, we clearly state:

“Until now, the XRD-CT temporal resolution has been considered its main drawback. However, in this work, we implemented a new data collection strategy, in principle applicable to all “pencil” beam tomographic techniques, where both tomographic axes (i.e. translation and rotation) are allowed to move simultaneously; this will be described in detail elsewhere. This new data collection strategy, coupled **with the brilliant X-rays produced at the ESRF and the state-of-the-art Pilatus2M CdTe area detector** allowed us to collect an XRD-CT dataset in less than 2 minutes (i.e. 117 s); the data collection rate is at least one order of magnitude faster than that previously reported.”

We have also explained why we performed the second experiment at ID15A – to decouple the spatiotemporal information observed during the reduction of the catalyst. In the main text, we mention (i.e. prior to Figure 5):

“However, the continuous growth of the Ni diffraction signal and decrease of the NiO and NiAl_2O_4 peaks (ca. $Q = 3 \text{ \AA}^{-1}$ and $Q = 2.55 - 2.65 \text{ \AA}^{-1}$ respectively) imply that NiO and NiAl_2O_4 were still present during the acquisition of this 3D-XRD-CT measurement (i.e. in several XRD-CT scans). In order to decouple these phenomena, we performed another diffraction experiment where we show that the NiO/ NiAl_2O_4 / Ni concentration gradients shown in Figure 4 are a purely temporal phenomenon (Figures S6 to S8 and accompanying text). The retardation in the full reduction of the Ni-O and Ni-Al-O species could be due to the formation/presence of water produced from the formation of metallic Ni closer to the reactor inlet.”

6) Experimental: The description of the reaction cells used is rather lacking in detail, and in the current form is probably not sufficient for most readers to understand the complexity of performing these measurements (e.g. specific geometry, positioning of gas blowers, mass spec, free rotation requirements). It is reasonable to assume that the cells used are the same as described in previous works by the authors [Ref 15 - O'Brien et al, Chemical Science 3, 509 (2012); Ref 19 - Senecal et al, ACS Catalysis 7, 2284-2293 (2017)] - while a full description

of the apparatus does not need to be repeated here, more appropriate citation in the experimental section would be appreciated. With this in mind, the publication of this manuscript is very likely to stimulate interest among the catalysis community to attempt similar experiments, likely with the same apparatus which is presumably available at ID31/ID15A. Hence, please provide an objective assessment of the drawbacks of the reaction cell used.

Specifically:

(i) it is not a 'closed' system, the top of the reactor is open to the environment, and backflow of ambient gas into the capillary can only be countered by supplying rather high flow rates of gas (in this work 100 ml/min during POX experiments).

(ii) the space velocities of the gases inserted into the system are therefore necessarily rather high. It would be useful to state the GHSV applied during tomography experiments, and how this compares to the laboratory experiments.

This is a very fair comment by the reviewer and we apologize for not providing this information previously. We have added a new section in the ESI ("Experimental setup at beamlines ID15A and ID31, ESRF") describing the experimental setups during the beamtime experiments, including photos of the various apparatus (Figures S1 and S2).

Backflow of ambient gas was not an issue in these experiments. We did not see any air signal in the mass spec data. The catalyst particles (and indeed the heating zone for the ID31 beamtime experiments when we used the Cyberstar hot air blowers) were positioned near the middle of the quartz tube, which is several cm away from reactor outlet (which is the exposed top end).

The GHSV is expected to affect the performance (CH_4 conversion and selectivity/yield for $\text{H}_2/\text{CO}/\text{CO}_2$) but the reaction products are not expected to change. We tried to replicate the POX XRD-CT experiment (i.e. from beamline ID31) in the lab by following the same experimental protocol. We could have explicitly mentioned the values for the GHSV but these can be calculated by dividing the total volumetric flow rate of the inlet gases over the mass of the catalyst bed (units of $\text{ml min}^{-1} \text{mg}^{-1}$). These values are provided in both the main paper and the ESI. Of course, the heating zone during the ID31 beamtime experiment was small (ca. 2 cm using the hot air blowers) and the quartz tube was also small (4 mm outer diameter quartz capillary as mentioned in the methods section of the paper). These were requirements for the *in situ* experiments as otherwise, the temperature gradients would become significant and the acquisition time would be a lot longer. These are well known requirements to the *operando* catalysis community and have been raised again and again in the past. We have also referenced review papers that discuss these things in detail.

Such requirements are not necessary for the lab catalytic experiments where the quartz tubes are significantly bigger (allowing for larger quantities of catalyst loading) and the temperature distribution is fairly uniform along as the reactor is placed inside a furnace. The aim of the laboratory test was to expose the catalyst to the same chemical environments as in the beamline experiment (i.e. keeping the sequence of the used gases the same and the concentration of the gases too). Of course, the conditions are never going to be identical but

the lab experiment serves to prove that this catalyst performs well under POX reaction conditions and that it starts deactivating when the CH₄:O₂ is 4:1. The XRD-CT data suggest that this is due to the formation and growth of graphite which is the key finding here.

7) Catalytic performance: The claim of an operando experiment suggested in the title rests entirely on the mass spectrometry data obtained, and specifically the observation of products (CO + H₂, byproducts CO₂, H₂O). This data is found only in the supporting information and is not completely convincing, for the following reasons:

(i) the $t = 0$ point is given as more or less the exact moment the gas conditions were changed from pre-reducing (20% H₂/He, 100 ml/min) to POX reaction conditions (30:4:1 He:CH₄:O₂, 105 ml/min).

(ii) The relative change in CO and H₂ mass traces before $t = 0$ (proof of product formation) is therefore difficult to see. The authors should clearly show the traces before POX conditions were introduced, so during the reduction step immediately prior.

(iii) Note further that the total flow rate of gas to the MS was not constant at this point (100 to 105 ml/min). The signal of ALL gases detected by the MS will therefore change.

(iv) the appearance of CO₂ in particular is rather apparent, but not totally convinced the POX reaction was occurring.

Linked to this previous point, the authors state in the supporting info (line 217-221) that all mass traces observed were stable during tomography studies, therefore catalyst deactivation was not expected. In Figure S13 I cannot see the trace of oxygen. I do not understand why CO₂ is zero and why the traces of water are not given. How does this reconcile with the gradual appearance of graphite, which is a likely product of combustion? Etc.

The mass spec data collected during the XRD-CT POX experiment were put in the ESI as they do not offer such significant information to be in the main paper. As we stated in the corresponding section in the ESI: “The mass spectrometry data acquired during the POX experiment are presented in Figure S11, where the signals from specific masses of interest are shown and serve to prove that the catalyst was captured in its active state during the POX reaction”.

The reviewer’s comment that upon switching to the POX reaction mixture the MS signal of all chemicals would change due to the 5 ml min⁻¹ difference in total flow rate is wrong. This assumption is wrong because the MS capillary is dragging with a constant flow rate of 20 ml min⁻¹. It is now clearly stated in the methods section. Maybe what the reviewer was trying to say is that the signal of all masses will change due to the fact that there is a different gas composition. For clarify, consider the simple examples:

If we send 30 ml min⁻¹ of 10 % H₂/He or 100 ml min⁻¹ of 10 % H₂/He, the signal that we see in the MS will be the same (as it is dragging with 20 ml min⁻¹).

If we are initially sending 30 ml min⁻¹ of 10 % H₂/He and then switch to 15 ml min⁻¹ of 10 % CH₄/He and 15 ml min⁻¹ of 10 % H₂/He, the MS signal of all masses will indeed change due to the fact that the concentration is different for all chemicals in the new mixture.

The absolute value of the various signals of the MS data presented in Figure S11 in the ESI do not have any meaning as the MS needs to be calibrated prior to the experiment but this was not possible during that beamtime experiment. The MS data presented here serve to prove that the catalyst was captured in its active state. The specific signal that proves this is actually the signal corresponding to H₂ where we can see that it significantly increases and is constant for the duration of the POX experiment. As mentioned in the methods section, the mass spec capillary is placed at the top of the catalyst bed. The presence of CO₂ and H₂O in the MS data could be due to unconverted CH₄ being burned to CO₂ and H₂O after the reactor outlet and the MS detecting these signals too.

For all these reasons (and because we agree with the reviewer's point of view), we performed the lab POX experiment trying to replicate as well as we can, the POX beamtime experiment. The quantitative results provided by the lab experiment under almost identical chemical environment prove that indeed we were able to capture the catalyst in its active state and also during deactivation.

8) Deepening the point of operando:

Since the word 'operando' appears in the title and abstract, this must be carefully re-considered. By (generally accepted) definition, an in situ study involves spectroscopic/microscopic/physical probing under operating conditions, while an operando study requires catalytic activity to be measured simultaneously so that structure-activity relations can be derived. Apart from the catalytic data discussed previously I have the following reservations:

(i) operando studies need to be done on a closed system - the tomography capillary setup used here is not a closed system, but open at the top.

(ii) while there may be some evidence of product formation (see discussion on MS above), there is no attempt at quantification, while even qualitative discussion of the MS data is relegated to the supplementary information.

To justify the use of the term 'operando', the catalytic activity data presented should be robust - otherwise this is an 'in situ' study with 'realistic' reaction conditions.

Here we strongly disagree with the reviewer. *Operando* does not imply quantitative analysis of the products and performance of the catalyst neither it needs to be a closed system. For a study to be claimed to be *operando* all that is needed is direct proof (with mass spectrometry, gas/liquid chromatography or IR analysis of the reactor outlet gas/liquid stream) that the reaction products are formed. The reviewer acknowledges the fact that we do provide such information with the mass spectrometry data presented in the paper. Furthermore, although we knew that we did not need to provide any further proof (i.e. quantitative analysis), we performed the lab experiments replicating the experimental protocol we followed during the synchrotron XRD-CT experiments providing quantitative results so we keep our claim that is in an *operando* study.

According to the website of the 4th International Congress on Operando spectroscopy (<https://workshops.ps.bnl.gov/?w=OperandoIV>) it says simply that: The study of catalysts in

their working state, the operando approach, has become a well-established and critical research area in catalysis. In this methodology the behavior of working catalysts is probed using physico-chemical methods while simultaneously measuring the catalytic activity and selectivity. According to the website of the 6th IC, it says: In operando studies, the behavior of working catalysts is probed using physico-chemical methods while simultaneously measuring the catalytic activity and product selectivity (<http://www.operandoconference.com/>). As regular attendees at such conferences, to the best of our knowledge, it has never been agreed that points i) and ii) should be used to define (in part) what does/doesn't constitute an operando study. This is most probably because it is difficult to mimic an industrial process using an operando setup and any hard definitions (if they could be agreed on) would risk marginalising a large percentage of the community.

This study would have been an *in situ* study if the catalyst was just exposed to a POX reaction mixture and we did not provide any evidence that the catalyst was actually “working”. It should be emphasized that this would have made the experimental setup a lot simpler but we believe the fact that we were able to capture MS data too during the XRD-CT experiment adds a lot of value to the study and makes it more relevant.

9) Figures: While the many interesting figures are enjoyable and well presented, emphasis should be laid on Figures 7 and 8. One may put some of them into the ESI. Figure 8 should be improved. I assume moving from left to right is the same slice measured at different times. Therefore please introduce a time scale on the x axis for the top part of the figure (what is the temporal resolution for these scans). Furthermore, the small graphs on the right have no axis labels. Please include labels on all figure axes. In the discussion on Figure 8 (line 313-318), the authors state that the decreasing intensity of the CeZrO₂ phases (10% in 3 hours) is ‘not very significant’, although without time scales on the graphs it is hardly to judge. The Pd intensity seems to change rather dramatically, could the authors comment on this?

We are glad that the reviewer liked our figures. Indeed, we spent much time making sure we only included in the main paper only those figures that were necessary to provide a flowing and comprehensible story. The 8 figures used is well within the limit for such articles in this journal. By the nature of multidimensional data, covering spatial, chemical and temporal/environmental conditions, it is not always easy to present material that is readily accessible to non-expert readers. Critical information to the materials under study, such as phase distribution, crystallite size and lattice parameters should naturally be included in such a study. We agree though with the reviewer that certainly figure 8 could be improved and consequently we have made the changes suggested by the author (added time axis at the bottom of the figure and x axis on the figures on the right side).

The Pd phase is a very minor one (Pd loading of only 0.2 wt. %). We are able to see a peak corresponding to this phase and the signal of this peak is decreasing as a function of time. We do not see any other diffraction peaks appearing though. We took the initiative and performed a high resolution XRD-CT measurement of a single catalyst particle and the results from the Rietveld analysis of this big dataset (640,000 diffraction patterns) are presented in Figures S8 and S9. In these data, we are able to extract diffraction patterns from the regions of high concentration of PdO and clearly show the PdO diffraction pattern. Unfortunately, this is not feasible with the lower resolution XRD-CT data. We are hoping in the future to conduct a single particle XRD-CT study with ultra-high spatial resolution (ca. 1 micron beam size) that

should be really able to provide us information about how the Pd is behaving under POX reaction conditions but with the resolution used in this study (ca. 30 microns), it is impossible to provide such information. Despite this limitation, we were able to track the solid-state evolution of the Pd containing phases showing that it is present as PdO in the fresh catalyst, it forms metallic Pd which is gradually converted to a Ni_xPd_y alloy and then gradually disappears under reaction conditions; these are all new observations and a technical accomplishment if one takes into account the low Pd content (0.2 wt. %) and the relatively large translation step size during the CT measurements.

Some minor comments:

- line 15, '5D' should be defined as '5-dimensional' on first use, this is not a common acronym

We performed this change.

- line 19-20, Angstrom and nm should have the word 'scale'

We performed this changes.

- line 33, "it is the brilliant X-rays generated at synchrotrons coupled with state of the art detectors and tomographic data collections that now allow acquisition of spatially-resolved signals under operating conditions". Not necessarily - it is very possible (99% of all such papers) to do spatially-resolved studies in 2D. Tomography is not required as this statement suggests (although 3D spatial resolution is always better than 2D). Please rewrite.

There seems to be a confusion here: a single XRD-CT "slice" (and every other "pencil beam" chemical CT technique) provide 2D spatial resolution, not 3D. 3D spatial information is provided if successive "slices" are collected at different z positions. Another way (which is probably what the reviewer is implying) is to perform a "mapping" imaging experiment. However, even in this case (which is indeed the vast majority of the papers as the reviewer mentioned), brilliant synchrotron-generated X-rays and state-of-the-art detectors are necessary in order to map a catalytic reactor (or any other functional material) relatively fast. As in this case, the important information would be derived by collecting successive (XRD) maps and investigate how the various gradients (e.g. physical, chemical, thermal) from the reactor inlet to the reactor outlet as a function of time under operating conditions. We have added modified the statement to make it clearer: "Non-destructive X-ray spectroscopic/scattering techniques are typically employed to study such materials but it is the brilliant X-rays generated at synchrotrons coupled with state-of-the-art detectors and tomographic data collections that now allow the acquisition of spatially-resolved signals from within the interiors of intact objects under operating conditions."

- line 43-44, "This is especially true for catalysts applied at the industrial scale where catalysts are needed to be produced on a scale where fine chemical control (and therefore sample homogeneity) is difficult to achieve". This sentence is difficult to read and the meaning is unclear.

Indeed it is and it was also not adding any value to the manuscript. As such we decided to remove it.

- line 65-66, "...XRD-CT temporal resolution has been considered its main drawback." What about the huge volumes of data obtained? Data storage and particularly big data processing is becoming a more relevant issue in modern catalysis and may to be mentioned.

This is very well spotted by the reviewer. Data handling (storage, transfer, processing) is actually the rate limiting step when one performs such experiments. We have now modified this paragraph to: "Until now, the XRD-CT temporal resolution has been considered its main drawback along with problems associated with large data sets and high volume processing. However, in this work, we implemented a new data collection strategy, the concept of which we introduced previously, in principle applicable to all "pencil" beam tomographic techniques, where both tomographic axes (i.e. translation and rotation) are allowed to move simultaneously; the technical details will be described in detail elsewhere. Also, we were able to meet the data handling challenge by analysing millions of diffraction patterns."

- line 76-78, the discussion of many dimensions (beyond 5D) is somewhat speculative and probably not necessary.

We disagree with the reviewer. Herein, we also show the potential of the technique and how it can be used to perform experiments that are even more complicated in the future. This information may prove to be useful to other researchers that would like to attempt something like that. However, we understand that the discussion regarding multi-dimensional diffraction imaging is not trivial and this is the reason why we have added a new section in the ESI discussing multi-dimensional chemical imaging.

- Methods - Catalyst preparation, were the catalysts pressed and sieved after synthesis? More information is required - this is also relevant to calculate whether the GHSV of the POX reaction was as expected for standard operating conditions. Without catalyst bed volume, the reader cannot derive the GHSV.

We believe that from a practical point of view, GHSV should always be provided in units of $\text{ml min}^{-1} \text{mg}^{-1}$. These can be calculated for every experiment (and indeed condition) provided in this work as we give the values for flow rates and catalyst mass (weight loading). Unfortunately, the way GHSV is reported in literature is extremely inconsistent. Regarding the catalyst preparation (this information was added to the method part of the paper too): The microspherical $(\gamma+\delta)\text{-Al}_2\text{O}_3$ with granules size of approximately $500 \mu\text{m}$ was used. The apparent density of catalyst is ca. 0.9 g/cm^3 . So if the catalyst loading was 35 mg (as it follows from the paper, part Method), then the catalyst bed volume was ca. 0.04 cm^3 . According to the Method part of paper, the flow rate of POX reaction mixture was 110 ml/min ($6600 \text{ cm}^3/\text{h}$). Therefore, the GHSV was $(6600 \text{ cm}^3/\text{h})/(0.04 \text{ cm}^3) = 165000 \text{ h}^{-1}$.

- Methods - Reactor cells, line 412, where exactly was the temperature measured? Top of the bed, bottom of the bed? Is it right to assume the temperature was not continually measured during reaction? How can we then be sure that the stated temperatures were actually experienced by the catalyst during in situ / operando studies, since there is no feedback control mechanism to maintain a steady temperature (e.g. modulations due to exothermicity)?

We performed temperature calibration prior to the in situ experiment. We have now added the temperature calibration curve in the ESI (Figure S3). The amount of catalyst used is small and the flow rates are high both of which were chosen so that the temperature fluctuation under POX would be minimal. It is possible to add a thermocouple to directly measure the

temperature in the catalyst bed but we try to avoid doing this on purpose. The reason being that for *in situ* experiments at synchrotrons typically small amount of catalysts loadings are used (in most cases due to restrictions related to the experimental setups) and as such the catalytic activity of a K type thermocouple (which is Ni-based) can be significant and can mess the MS data. Furthermore, the diffraction signal from the thermocouple can be potentially be very strong and may even damage the detector (i.e. there should be a maximum photon count rate of 1,000,000 photons s⁻¹ for the Pilatus2M CdTe detector). As such we always choose to waste precious beamtime and perform a temperature calibration experiment (so the reactor system heated up to the operating temperature) without beam, before we perform the *in situ/operando* XRD-CT experiments. In future experiments, we will try to use also an IR camera during the XRD-CT experiments to capture any potential temperature gradients along the catalyst bed (although this will make the experimental setup even more challenging).

- Methods, line 416, 'tomographic' is misspelled.

We performed this change.

- line 437, "An XRD-CT was at the middle of the catalyst bed was collected at ambient conditions." Please rewrite.

We performed this change

- line 443, 'successive' is misspelled.

We performed this change

- line 443-444, "Eight successive XRD maps were collected covering the whole bed..." This implies that the authors actually scanned from top to bottom every single part of the catalyst bed. With a 0.5 mm vertical step size this is surely not the case? Rather selected slices must have been scanned?

It is correct. As mentioned in the ESI, the total length of the catalyst bed was 5.5 mm.

- line 462, problem with reference formatting.

We performed this change

Reviewer 3:

Tracking the chemical evolution of catalysts under reaction conditions is important for better understanding of the structure-function relationships. In this manuscript, a novel 5D tomographic diffraction imaging technique was applied in practical chemical experiment (POM), which enabled the authors to extract heterogeneities in the catalyst under different chemical environments. The results are quite interesting. However, some key issues concerning this technique should be emphasized and additional data need to be provided. I would recommend that publication is withheld until following information is supplied.

We thank the reviewer for acknowledging the state-of-the-art work presented in the paper and that in order to gain an understanding of the structure-function relationships in complex, real

life (i.e. rather than idealised model powders), catalytic systems, it is vital to effectively tracking their evolving chemistry under real process conditions.

1. It is generally accepted that the nanometer size of supported metal catalysts is closely related with its catalytic performance. As for XRD-CT technique, the limit of crystallite size resolution should be given in the manuscript, since the size of Pd/PdO is not showed (Figure 1).

We have performed a high resolution XRD-CT scan of a single catalyst particle with a $1\ \mu\text{m} \times 1\ \mu\text{m}$ focused beam. We also performed Rietveld analysis of these data (640,000 diffraction patterns) and are able to show that there are regions of high concentration of PdO. This was also implied by the lower resolution 3D-XRD-CT data but the high resolution ones clearly demonstrate this. This means that the PdO phase is not well-distributed over the catalyst particles. As it can be derived from the results of these two datasets (i.e. low and high resolution scans), it is not trivial to strictly define what is the limit of crystallite size resolution for the XRD-CT technique and it is also beyond the scope of this work. In fact, we have previously shown that XRD-CT can detect crystallite sizes down to a couple of nm (S.D.M. Jacques *et al.*, Nat. Comm., 4, 2536, 2013). It should be emphasized though that the whole point of performing chemical CT measurements of complex heterogeneous materials systems is also to gain an understanding of how the various components are radially distributed. Such knowledge can help researchers understand how a system can be later finessed and indeed can lead to the rational design of improved materials.

2. The components of supported catalyst are complicated. The signals from different phases or metals maybe overlap with each other and affect the accuracy of data analysis. The spatial resolution of phase distribution maps should be emphasized.

We thank the reviewer for acknowledging that this was a very challenging system to investigate (e.g. in contrast to simpler catalytic systems where there is only one metal (e.g. Pt) supported (e.g. Al_2O_3)). As we mention, several times throughout the paper, we performed Rietveld analysis of the spatially-resolved diffraction patterns in each XRD-CT dataset, leading to the Rietveld analysis of ca. 2,000,000 diffraction patterns in total. The Rietveld method [J. Appl. Cryst. (1969). 2, 65-71] has been used for the past ca. 50 years to analyse diffraction patterns of multi-component polycrystalline samples. This approach has also been used in other recent XRD-CT studies (e.g. Sottmann J. *et al.*, Angewandte Chemie International Edition 56, 11385-11389, 2017). For example, we show in Figure S5 in the ESI, the result from the Rietveld analysis of an XRD-CT derived diffraction pattern and it can be clearly seen that fit is of excellent quality. However, in order to further support the XRD-CT results, we performed SEM/EDX measurements of the fresh and spent catalyst (Figures S21 to S24).

3. The $\text{CeO}_2\text{-ZrO}_2/\text{Al}_2\text{O}_3$ support was prepared by co-impregnation method. It is reasonable to imagine that the Ce/Zr species would load on the surface of Al_2O_3 particle. However, obvious CeO_2 signal was found at the core of Al_2O_3 particle. The inherent reasons should be discussed (or only due to overlap of 3D signal?).

This is not due to overlap, it is real observation as clearly shown in Figure S3 of the ESI. As mentioned previously, the Rietveld method can resolve such peak overlapping problems (but it should also be mentioned that the $\text{CeO}_2\text{-ZrO}_2$ peaks are not overlapping – this is also shown in Figure S3 of the ESI). This is how we were able to discriminate between four different $\text{Ce}_x\text{Zr}_{1-x}\text{O}_2$ crystalline species present in the catalyst particles. The reality is that the preparation method will dictate the spatial distributions but despite a relatively simple procedure there will be several factors at play such as the acidity of the catalysts, the local pH during wetting and drying steps, the local concentration of components etc. It would indeed be an interesting study to look at how different preparation protocols affect the distribution of the components and try and relate these to performance. We had to limit our study and focussed alone upon the best performing catalyst. The previously discussed results were verified by the ultra-high resolution XRD-CT scan (1 micron step) as shown in Figures S8 and S9. We have also performed SEM/EDX measurements of the fresh catalyst which confirm the presence of Zr species mainly near the surface of the catalyst particles while Ce species are seen to be present everywhere (Figure S21 to S24).

4. In this manuscript, the chemical evolution of Ni species was detailed studied while that of Pd species, another active component, was seldom investigated. The authors should explain the role of Pd in the POM reactions.

We disagree with the reviewer that we did not discuss this. We were able to track the solid-state evolution of the Pd containing phases showing that it is present as PdO in the fresh catalyst, it forms metallic Pd which is gradually converted to a Ni_xPd_y alloy and then gradually disappears under reaction conditions; these are all new observations and a technical accomplishment if one takes into account the low Pd content (0.2 wt. %) and the relatively large translation step size during the CT measurements. It should be emphasized that Pd is a strong scatterer but at 0.2% will be difficult to see using conventional XRD. On the other hand, with the local signals present in a reconstructed XRD-CT dataset, we are able to see a peak corresponding to this phase and the signal of this peak is decreasing as a function of time. We do not see any other diffraction peaks appearing though. Although at first glance, this may not look as important as the information we reveal for the Ni species, this is not true. The reason is that the Pd-containing phases cannot be seen at all using conventional lab XRD instruments (Ismagilov IZ, *et al. International Journal of Hydrogen Energy* **39**, 20992-21006 (2014), Ismagilov IZ, *et al. Kinetics and Catalysis* **56**, 394-402 (2015)) while with the synchrotron XRD-CT that yields local signals we were able to map the Pd-containing species (although challenging) and also see them evolving under different operating conditions. We are hoping in the future to conduct a single particle XRD-CT study with ultra-high spatial resolution (ca. 1 micron beam size) that should provide us more information about how the Pd is behaving under POX reaction conditions but with the resolution used in this study (ca. 30 microns), it is impossible to provide such information.

5. The TG experiment should be conducted for the catalysts after POM reaction (3h) to further confirm the amount of coke.

We have performed TGA measurement of the spent catalyst as requested by the reviewer and added the results in the ESI (Figure S25). The results serve to very clearly show that there is one type of coke forming in the catalyst during the *in situ* experiments which is the graphitic

carbon as identified from XRD-CT. We thank the reviewer for requesting this extra measurement.

6. As the author indicated, the 5D tomographic operando diffraction imaging showed high spatial and temporal resolution with good element sensitivity, a more detailed discussion on the coke deposition process (where and when it starts? On Ni or Pd particle or interface?) should be provided to strengthen the confidence on this novel technique.

We agree with the reviewer's comment and indeed we added more text regarding that in the main text of the paper. In short, we observe the Ni metal and graphite phase distribution maps are directly related (no correlation with Pd). This is though a first experiment (it is a communication paper after all) and we report the first ever XRD-CT data of a working POX catalyst. We hope that we will have the chance in future beamtime experiments to investigate the catalyst under POX conditions focusing only on the coke formation. However, in this work we also showed, for the first time, what is the state of the activated catalyst, after re-oxidation, the solid-state evolution of the crystalline Ni-containing phases (i.e. $\text{NiAl}_2\text{O}_4/\text{NiO}/\text{Ni}$) during redox and the multiple roles of the $\text{CeO}_2\text{-ZrO}_2$ promoters. These are all new findings and we expect that the catalysis community will appreciate them.

7. The construction of structure-function relationship not only lies in recognition of the particle size or phase composition but also the electronic property of active species. The additional information from other characterization techniques such as XPS or XAFS except the 5D tomographic diffraction imaging were hoped to be given and the connection of these results can be discussed.

Here we disagree with the reviewer for several reasons. First, the crystal structures observed are implicit to the electronic state of the elements present. Then, the work presented in this paper clearly demonstrates that there are physico-chemical gradients taking place in the catalyst both radially and axially under different operating conditions (we even show how the complex the distribution of the different phases are in the fresh catalyst). This information can be obtained in a non-destructive manner only by performing a chemical CT experiment (and more specifically XRD-CT in order to differentiate between the various crystalline species present in the catalyst). XPS and XAFS point measurements, although welcome, would offer little to any more extra information about the system under study. It should finally be noted that XPS is an *ex situ* technique and XAFS (XANES/EXAFS) imposes severe limitation for the reactor thickness if one wants to probe the Ni or Pd metal. For these reasons, this experiment cannot even be conducted for XPS and XAFS ($T = 800\text{ }^\circ\text{C}$, flow of several gases at atmospheric pressure and reactor thickness of several mm).

8. The figure caption should be more intelligible. Taking Figure 8 as an example, the meaning of different color and time schedule of different pictures should be provided.

Regarding Figure 8, we agree with the reviewer that it has to be improved: we have now added a time axis at the bottom of the figure and labels to the figures on the right side of Figure 8.

Reviewer #1 (Remarks to the Author):

In the first manuscript, it was unclear "5D" imaging but the authors revised their manuscript in line of the comments of the reviewers. The additional data requested are also provided in ESI. The present manuscript would be accepted for publication.

Minor revision:

The authors claimed that all figures are necessary for their discussion, but it is difficult to look several figures visualized same parts in the catalysts e.g. Figure 2 and Figure 3. They can combine them in a figure and discuss correlation between these parameters.

Imaging of coke formation should be measured and reported in their next paper.

Reviewer #3 (Remarks to the Author):

The authors have responded to most of my questions or suggestions. There is no doubt that 5D tomographic diffraction imaging technique can give a good supplement for the existing characterization techniques. Compared with conventional TEM/SEM or XRD, it can give more specific spatial distributions of particle sites and track chemical evolution of catalysts under reaction condition, which is a good advance. However, the multiple characterizations were helpful for heterogeneous catalysis, particularly to establish the relationship between structure and activity. As to the structure, the influences come from not only the particle size, particle morphology, the phases, but also more importantly, the electronic property of active sites. During the POM to syngas, the active sites can occur in various electronic states, for example, metallic species, metal oxides and also metal with slight positive valence. In this case, the authors give too many rebuttals and do not answer some questions directly. Such as on question 7, the author thought that "XPS and XAFS point measurements, although welcome, would offer little to any more extra information about the system under study.", which is wrong in my opinion. In situ or quasi in situ XPS or XAFS also have developed for several years and can give useful information about the electronic property and the coordination of a metal, even under the reaction conditions. The authors were hoped to give more experiments and more discussions about the technical supplement for heterogeneous catalysis. I do not suggest the publication in current form for the wide readership of Nat. Commun.

Andrew Beale
UCL Chemistry/UK Catalysis Hub
Rutherford Appleton Laboratory
R92
Harwell Oxford, Didcot
OX11 0FA
United Kingdom
Email: Andrew.Beale@ucl.ac.uk
Phone: 00 44 (0) 1235 567842

14th September 2018

To the editor for the Nature Communications,

Dear Dr. Long Chen,

On behalf of my co-authors and myself, I wish to thank you for considering the following manuscript as an article in Nature Communications, subject to minor revisions.

We have added a “Data availability” section in the main paper according to the journal regulations.

The comments submitted by the reviewers have been addressed with the following changes where necessary:

Reviewer 1:

In the first manuscript, it was unclear “5D” imaging but the authors revised their manuscript in line of the comments of the reviewers. The additional data requested are also provided in ESI. The present manuscript would be accepted for publication.

Minor revision:

The authors claimed that all figures are necessary for their discussion, but it is difficult to look several figures visualized same parts in the catalysts e.g. Figure 2 and Figure 3. They can combine them in a figure and discuss correlation between these parameters.

Imaging of coke formation should be measured and reported in their next paper.

We would like to thank the reviewer for accepting the revised version of the manuscript for publication. At the reviewer’s request, we have combined Figures 2 and 3 in one larger Figure. We agree with the reviewer’s assertion that it is indeed now easier for the readers to appreciate the various heterogeneities in 3D. We appreciate the suggestion of the reviewer and we plan to focus on this aspect in future experiments.

Reviewer 3:

The authors have responded to most of my questions or suggestions. There is no doubt that 5D tomographic diffraction imaging technique can give a good supplement for the existing characterization techniques. Compared with conventional TEM/SEM or XRD, it can give more

specific spatial distributions of particle sites and track chemical evolution of catalysts under reaction condition, which is a good advance. However, the multiple characterizations were helpful for heterogeneous catalysis, particularly to establish the relationship between structure and activity. As to the structure, the influences come from not only the particle size, particle morphology, the phases, but also more importantly, the electronic property of active sites. During the POM to syngas, the active sites can occur in various electronic states, for example, metallic species, metal oxides and also metal with slight positive valence. In this case, the authors give too many rebuttals and do not answer some questions directly. Such as on question 7, the author thought that “XPS and XAFS point measurements, although welcome, would offer little to any more extra information about the system under study.”, which is wrong in my opinion. In situ or quasi in situ XPS or XAFS also have developed for several years and can give useful information about the electronic property and the coordination of a metal, even under the reaction conditions. The authors were hoped to give more experiments and more discussions about the technical supplement for heterogeneous catalysis. I do not suggest the publication in current form for the wide readership of Nat. Commun.

As remarked by reviewer 1, we have provided a lot of new and additional data that were requested by all reviewers. We also expended considerable effort in performing a plethora of additional measurements that were not suggested by the reviewers. These include high resolution XRD-CT (1 micron beam) of a single catalyst particle, 3D-XRD-CT and micro-CT of a fresh catalyst bed, SEM and EDX of the fresh and spent catalyst and TGA of the spent catalyst. We have also added a section in the ESI discussing multi-dimensional chemical imaging and also a short literature review showing the very important technical achievement we performed regarding the data collection strategy.

With regards to the only question highlighted from the previous review, question 7, concerning ‘The additional information from other characterization techniques such as XPS or XAFS.... were hoped to be given and the connection of these results can be discussed’. Unfortunately, in the original question there was no remark of how these measurements should be performed to give the added value. Should this have been on the fresh catalyst, the reduced one, the reacted one or for the measurements to be performed under reaction conditions? No context was given. The problem as we see/saw it is how can we perform such XPS/XAFS measurements so as to provide *meaningful* additional information? In the first instance our study is an *operando* study; the salient points here being that the reaction is carried out and that actual catalytic conversion is determined. There are four main reasons then why performing *operando* XPS experiments will struggle to provide this meaningful additional information:

- 1) Currently there is no reaction chamber that we are aware of and that is readily accesible and that can reach the reaction temperatures of 800 °C used here.
- 2) It is not possible to measure powders or structured catalysts as we have measured here.
- 3) Such facilities cannot work at the gas pressures and space veolcities employed here.
- 4) The large dead volume in such reaction chambers coupled with the small amounts of sample present mean that it is not possible to measure catalytic activity.

As such the only way we could measure and perform a quasi *in situ* XPS measurement would be to grind the catalyst down to a very fine powder and press it as a pellet or else disperse it over an XPS sample holder and heat it to below the temperature of reaction in a gas atmosphere 10/20 % of what was used in the XRD-CT study. So whilst we agree with the referee these studies would be revealing, the deviations away from the experiments performed in the XRD-CT study would be so

great so as to render such data of limited use. In many ways if we were to draw parallels from such two very different studies this would also invite comments from catalysis experts as to how relevant the connection was when two samples were measured under two very different conditions.

For XAFS, whilst we agree that it is possible to measure under conditions more akin to those used in our XRD-CT study, there are still challenges to overcome to be able to acquire data that you could use to relate the two studies. The most relevant manner in which to do this would be to perform something like XAFS-CT. However, it is not technologically feasible to perform XAFS-CT at reasonable time scales for this sort of catalysis experiment since there are multiple edges to measure at multiple energies. It would therefore be extremely slow (i.e. hours to days). Furthermore beam attenuation at the absorption edge of interest severely limits the size of reactors that can be probed and is technically very challenging since this requirement will vary depending on the element being studied. For example one could use a comparatively wider/thicker reactor for examining Ce/Zr and Pd but it would likely be too absorbing for studying Ni. Conversely one would then have to use a comparatively narrower reactor for. However, there would likely be too little Pd in the reactor to obtain a reasonable XAFS signal. Hence two reactors would be needed.

This leaves XANES/EXAFS point measurements. Firstly since there are four edges (Pd, Ni, Ce, Zr) each one can be considered a separate experiment, particularly since the Pd is present in low amounts requiring a different setup (i.e. fluorescence instead of transmission). With the repeating of experiments there is always a risk that the conditions are not directly comparable since fresh catalyst would have to be added each time. Furthermore we show clear evidence of axial gradients in our reactor which would require mapping experiments be performed, but even then that would not capture the radial information provided by XRD-CT. The impact/importance of this is nicely demonstrated in this current work where we collected XRD maps (ESI) and if compared with the XRD-CT data it can be clearly seen that the amount of information is much smaller. As an example, with the use of XRD-CT, we were able to differentiate between four different CeO₂-ZrO₂ species present in the catalyst particles. We were also able to track evolving solid-state chemistry of the active catalyst components, the Ni and Pd-containing species. We captured the Ni evolving from NiO to NiAl₂O₄ and to metallic Ni under different imposed chemical environments while the Pd species start as PdO and then evolve to Pd and eventually form NiPd alloys. Single point XAFS is well-known to struggle with multicomponent local environments (i.e. sample heterogeneity) and hence it would be very difficult to obtain any new additional insight (i.e. charged metal surface) against the backdrop of these multiple phases. We were also able to show that there is carbon forming under harsh POX conditions and identify it as graphite and show that the distribution of this phase is directly correlated to the Ni one; XAFS would struggle to do this as a single point measurement (particularly considering the issue of temperature effects on σ^2 which would dramatically reduce the signal amplitude, particularly for elements at low loadings to likely render any data useless). All these results were presented in spatial maps where we showed that different phases followed many different distributions (e.g. some egg yolk, others egg-shell and others uniform distribution). All these results would have not been feasible to obtain without the tomography aspect of the measurement (e.g. this result cannot be extracted from the XRD mapping data).

We agree with the reviewer that XPS and XAFS are powerful techniques revealing insight at a more local level than what XRD does. In fact, several of the co-authors of this manuscript are experienced synchrotron users working with spectroscopic techniques and specifically with XANES/EXAFS. We also agree that over the past decades there has been significant progress with

the XPS and EXAFS techniques and these are expected to continue to develop for the following decades too and that these developments will be important for future catalyst development studies.

In this work, we performed the first 5D tomographic diffraction imaging experiment looking at the solid-state chemical evolution of a complex Ni-Pd/CeO₂-ZrO₂/Al₂O₃ catalyst under various operating conditions. For example, we managed to show the evolution in 3D of the Ni species (Figure 6) as a function of the imposed chemical environment; this Figure alone represents a major step forward in the characterisation of functional materials under real process conditions in general. Indeed the reviewer acknowledges that “Tracking the chemical evolution of catalysts under reaction conditions is important for better understanding of the structure-function relationships. In this manuscript, a novel 5D tomographic diffraction imaging technique was applied in practical chemical experiment (POM), which enabled the authors to extract heterogeneities in the catalyst under different chemical environments. We are interested in studying other aspects of the specific system in the future (e.g. a more thorough investigation of the coking process as suggested by reviewer 1) but it is unreasonable to resolve every chemical aspect of such a complex and evolving materials system with one study. We believe though that, with this 5D tomographic diffraction imaging study, we have provided a new insight into this system and have paved the way to design improved catalysts.

We hope therefore to have tackled the comments/queries raised by the referees and we look forward to hearing the outcome of your deliberation in due course.

Yours Sincerely,

Andrew Beale